# LiteFeatNet: A parameter-efficient and performance-centric deep learning model for multi-ocular disease identification using intermediate feature reduction from fundus images

Usman Rafi[1]*, Qamar Nawaz[1]*, Muhammad Ahsan Latif[1], Aisha Khatoon[2]

**1** Department of Computer Science, Faculty of Sciences, University of Agriculture, Faisalabad, Punjab, Pakistan, **2** Department of Pathology, Faculty of Veterinary Science, University of Agriculture, Faisalabad, Punjab, Pakistan

* usmanrafiuafcs@gmail.com (UR); qamar@uaf.edu.pk (QN)

## Abstract

Convolutional Neural Networks (CNNs) require a larger amount of input samples and computing resources to learn discriminative features for accurate identification of multiple retinal conditions, making the development and deployment of such models challenging on limited computing resources. This study presents a robust CNN (called LiteFeatNet) that requires fewer trainable parameters, computational resources, and processing time for accurate prediction. To enhance robustness and reduce computational time, a pre-trained NASNetMobile backbone is employed, and a method for time-efficient discriminative feature extraction from deep intermediate layers is proposed. The extracted features are refined using a spatially-aware feature map reduction module and classified using a custom classification module with fewer number of trainable parameters, reduced computational resource requirements, and computational time. Experiments are conducted using 1824 images from three distinct class labels in the Retinal Fundus Multi-Disease Image Dataset (RFMiD), with a 60:20:20 train-validation-test split. The LiteFeatNet architecture has a compact size (19.87 MB) and was trained using a standard pre-processing pipeline and training configurations. It outperformed twelve state-of-the-art models, achieving the highest testing accuracy of 90.33%, precision of 90.69%, recall of 90.33%, and F1-score of 90.27%, with a fast, impressive inference time of 4 milliseconds per image. Further, a generalizability study was also conducted using an external dataset, RFMiD 2.0, and the LiteFeatNet achieved competitive performance with quicker testing time compared to other architectures. To evaluate the scalability and adaptability of our proposed integrated framework for larger multi-class problems, we assessed the scalability by increasing class label complexity using two additional disease categories. Results validated the effectiveness and computational

**Data availability statement:** All three datasets can be found at the following recommended repositories. Dataset1 - RFMiD 1: https://www.kaggle.com/datasets/andrewmvd/retinal-disease-classification Dataset2 - RFMiD 2.0: https://zenodo.org/records/7505822 Dataset3 - Eye Disease Dataset https://www.kaggle.com/datasets/linabennaa/eye-disease-image-dataset-mendeley The code developed for this study is made publicly available on GitHub. The code can be accessed at the following URL: https://github.com/usmanrafics/LTFTNET.

**Funding:** The author(s) received no specific funding for this work.

**Competing interests:** The authors have no competing interests to declare that are relevant to the content of this article.

efficiency of this integrated framework compared with 9 baseline architectures. An ablation study was also conducted using the LiteFeatNet and two top-performing transfer learning architectures to validate that the synergistic combination of deep feature extraction and feature map refinement is the primary design decision behind the success of the LiteFeatNet architecture. The evaluation metrics, thus obtained, strongly suggest that the proposed LiteFeatNet is lightweight, fast, and robust, rendering it suitable for deployment in low-resource clinical settings.

## 1. Introduction

Among the six senses, human vision is of critical importance, as most real-world data is acquired and processed through the human vision system for understanding and interpretation. The human vision system comprises the eyes, optic nerve, and the brain, which work together in a complex yet understandable manner. Among the various parts of the human eye, the retina is a key component that serves as a photoreceptor, converting light signals into electrical pulses that are carried to the brain by the optic nerve for interpretation. Among various retinal diseases, Diabetic Retinopathy (DR), being common and progressive, is one of the leading causes of mild to severe vision impairments that may lead to irreversible damage as well as blindness if left untreated. Media Haze (MH) causes opacities in the eye's optical media leading to blurry vision, and it is also quite common worldwide. Cost-effective, non-invasive fundus imaging has enabled the early detection and effective management of patients. In fundus imaging, a specialized camera captures a colored image of the retina. Fundus images can show retinal veins, arteries, fovea, macula, optic disc, and optic nerve. Fig 1 shows a typical fundus image.

The fundus camera is not self-diagnosing, and human intervention is still required to identify and interpret abnormalities (if present). Human intervention is not straightforward at all, as a single disease may manifest in different patterns on fundus images across different individuals. Additionally, multiple diseases may present as overlapping patterns, which can sometimes lead to confusion and potentially misdiagnosis. Considering the mentioned facts, accurate diagnosis is crucial for patient care. However, in low-resource and underdeveloped countries, particularly in Asia, the availability of skilled ophthalmologists and ample consultation time is constrained due to the rapidly rising number of patients suffering from multiple retinal diseases [1].

Consequently, ophthalmologists, in particular, are taking a keen interest in utilizing Artificial Intelligence (AI)-assisted automated retinal disease diagnostic tools for providing the necessary patient care. Integrating AI-assisted Computer-Aided Diagnosis (CAD) systems benefit healthcare system in numerous ways by reducing diagnostic errors, improving the speed of diagnosis, and educating young ophthalmologists [2]. Therefore, to be more effective, these tools must be both accurate and computationally efficient, enabling their development and deployment on low-powered hardware such as smartphones, embedded systems, and even low-resource computers.

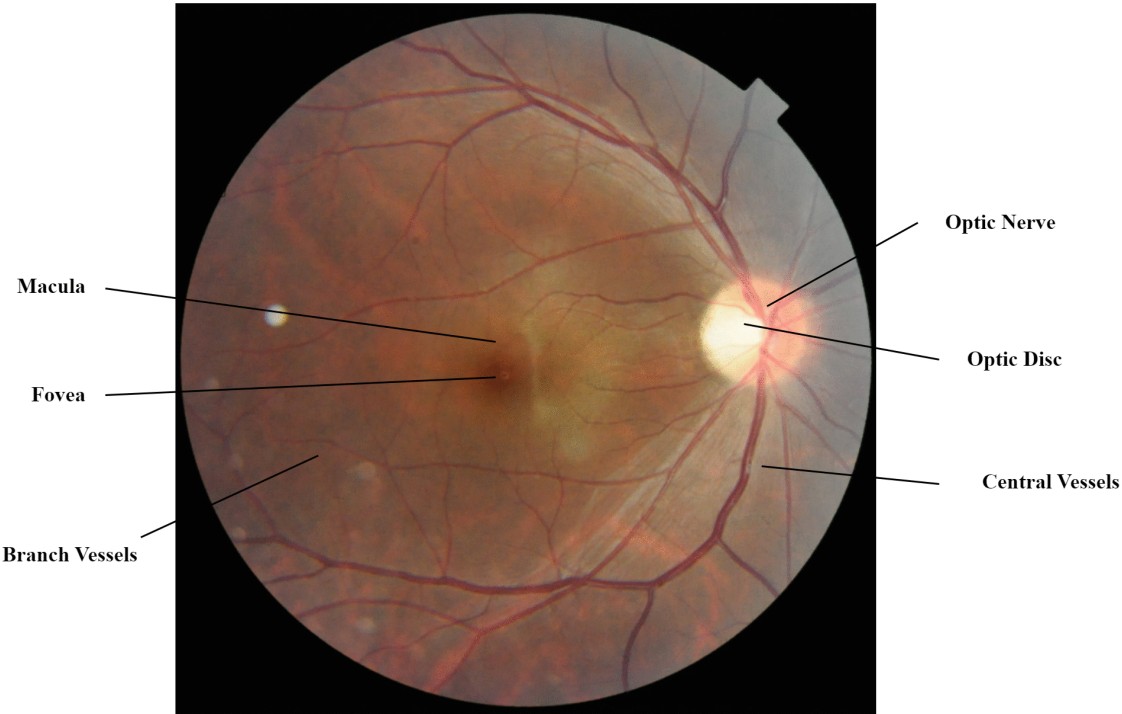

**Fig 1. A typical colored fundus image.**

Convolutional Neural Networks (CNNs) are among the most widely adopted AI tools for geospatial, precision agriculture, and medical image analysis, supporting a variety of tasks [3]. To overcome the numerous challenges faced by DR screening programs, [4] introduced Smartphone-Based Fundus Imaging (SBFI) and automated diagnosis using transfer learning with a fine-tuned EfficientNetB0, due to its resource efficiency. Numerous pre-trained and customized models have demonstrated superior performance in detecting and identifying various retinal diseases. However, to achieve high performance, CNN architectures rely on a large numbers of images and millions of parameters, and require intensive computing resources for development and deployment, which reduces their suitability for the development and real time usage on low-resource computing devices [5]. Additionally, not every transfer learning technique is lightweight and capable of capturing all disease-relevant features, thereby making the reliability of diagnosis questionable. Therefore, there is a need to develop an efficient, performance-centric, and robust disease identification model that requires minimal training and computational resources, to ensure it can be deployed in resource-constrained environments. Since 2022, significant research has been conducted on building lightweight deep learning models, particularly for edge computing [6]. A recent study introduced a cost-effective, scalable model (called Low-Cost CNN) for identifying four retinal conditions in resource-scarce environments. The system included numerous hardware components (magnification lens, camera, and Raspberry Pi) for capturing and processing fundus images, and a custom CNN-assisted web-based user interface hosted on Amazon Web Services (AWS) for detection and identification [7].

In this study, a performance-centric, computationally efficient, and robust deep learning-based integrated model, called LiteFeatNet, is proposed that leverages transfer learning for accurate diagnosis of multiple retinal conditions. Unlike conventional transfer learning setups, intermediate-level disease-rich discriminative features were extracted from the backbone using Deep Intermediate Feature Extraction (DIFE), refined using a Pointwise Feature Map Reduction (PFMR) module, and fed to a Lightweight Custom Classification (LCC) module for predictions. The LiteFeatNet architecture

achieved the highest accuracy with fewer trainable parameters, a smaller model size, and computationally efficient deployment. Moreover, it has also outperformed numerous architectures, including 6 cutting-edge transfer learning models: DenseNet121, MobileNet, MobileNetV2, NASNetMobile, Swin Transformer V2-Small, and Swin Transformer V2-Tiny, while achieving faster computation. Generalization (on an external dataset), scalability assessment, and ablation studies also validate its suitability for deployment in resource-constrained clinical environments such as portable retinal scanning systems and mid-range smartphones. Such environments typically have limited memory, making it difficult for AI-assisted inference to run smoothly. Usage of compact models such as LiteFeatNet makes model storage and loading easier, enabling reliable, memory-efficient, and fast inference.

### 1.1. Research contributions

The main contributions are:

1) Design and develop a lightweight, computationally-efficient, and reliable deep learning-based integrated architectural framework for the accurate identification of multiple retinal conditions.

2) Demonstrate that DIFE-based intermediate features, when processed through the PFMR module, achieve superior performance and reduce computational resource requirements in comparison to full-layered feature extraction from the pre-trained CNNs.

3) Validate the superior performance of the proposed integrated architectural framework by comparing it with various benchmark architectures.

4) Demonstrate the generalization and robustness of LiteFeatNet using an external dataset.

5) Evaluate the scalability, adaptability, and robustness of architectural components for larger multi-class problems.

6) Endorse the effectiveness and computational efficiency of DIFE and the PFMR module through a comprehensive ablation study.

The rest of the article is divided into numerous sections and sub-sections. Section 2 presents a critical review of the latest and robust models for image classification, particularly retinal disease classification. Section 3 and its sub-sections clearly define the overall research pipeline. This section is further divided into sub-sections that clearly define every step of the research framework both theoretically and diagrammatically, mathematically and algorithmically. It also outlines the experimental setup, including the training hyper parameters and hardware specifications. Section 4 highlights and discusses the promising results achieved by the LiteFeatNet architecture. Sections 5, 6, and 7 present well-structured generalizability, scalability assessment, and ablation studies of the proposed model, whereas Section 8 showcases the conclusion, limitations, and future work of this study.

## 2. Brief overview of existing literature

There are numerous ocular conditions responsible for mild to severe vision impairments, each associated with different pathological causes, including hypertension and diabetes. In 1990, diabetes was found to be the underlying cause of one per cent of cases of blindness, while in 2017 it jumped to 4 per cent. According to well-researched forecasts, the number of cases of vision loss is estimated to increase rapidly by 2025. To counter vision impairment cases and reduce the number of years lived with the disease, urgent measures are required [8]. This ever-growing patient population, particularly in developing and resource-constrained regions, requires the implementation of innovative, reliable, and scalable solutions to support clinical decision making. In this regard, AI has opened new horizons for decision-making, with its wide range of applications including precision agriculture, the automobile industry, and robotics. AI-assisted decision-making is unbiased, enabling patients to receive appropriate treatment, making it suitable for integration into the healthcare system.

Moreover, AI can reduce stress among medical practitioners, thereby improving decision-making. However, the provision of accurate and sufficient amount of data is highly critical to the successful implementation and effective operations of AI-assisted disease diagnosis systems [9].

As a subfield of AI, deep learning has become very popular for its ability to handle complex data with many features. Deep learning models need thousands of samples to become robust in classifying DR and other medical pathologies. Processing such a large number of features requires substantial computational resources and time. However, due to the limited availability of larger datasets, transfer learning is employed to utilize pre-learned features, thereby reducing computational overhead and the requirement of extensive model training [10]. In order to facilitate patients in remote areas, [11] studied various transfer learning models for the detection of DR and identification of DR grades. For all architectures, training was performed for 100 epochs, using a batch size of 64 on images from multiple datasets. VGGNet achieved the highest accuracy; however, it used over 11.8 million parameters, making it a resource-intensive one.

Melanoma has unique characteristics and skin appearance patterns. So, three pre-trained models, VGG19, ResNet18, and MobileNetV2, were utilized in three different ways for performance assessment. Experiments revealed that a hybrid deep learning-based feature extractor and a Machine Learning (ML) based classifier achieved an accuracy of 92.87%. Features extracted from ResNet18 and MobileNetV2 backbones were fused and fed to the Support Vector Machines (SVM) classifier [12]. The inclusion of the MobileNetV2 architecture yielded a lightweight feature extractor and a classifier.

Several studies were also conducted to develop lightweight and parameter-efficient deep learning models for identification of plant diseases. To enable quick, real-time detection of pepper disease using smartphones during the cultivation season, [13] introduced a novel multiscale feature extractor and customized classification block based on the traditional VGG architecture due to its robust feature extraction performance. The modified VGG was lightweight, achieving a 98% parameter reduction without affecting the predictive performance. In a study, [14] proposed a hybrid and lightweight plant disease detection model, featuring a 16-layered CNN-based feature-extractor and ML-based SVM classifier. The feature-extractor model had only 0.39 million parameters. In another study, [15] introduced a 16-layer lightweight CNN (CNN-2) for classifying four different categories of retinal pathologies. The model underwent training for 100 epochs with a 60:40 train-test split.

Similarly, for the classification of Normal (NL), MH, and DR, [16] introduced two lightweight transfer learning architectures that utilize the pre-trained MobileNetV2 and NASNetMobile (termed as LightCNN) as backbone feature extractors with 5 layers in the classification module. For both architectures, training process ran independently for 30 epochs with a batch size of 32, using an augmented dataset having 224x224x3 images. The architecture achieved a test accuracy of 89.50%. Three different custom CNN architectures were also introduced and compared for the classification of four different retinal conditions. Experiments were performed both with and without data augmentation. The 13-layered CNN (termed as CNN-1), when trained for 20 epochs using the Adam optimizer, exhibited a smooth training behavior and achieved accuracies of 91.94%, 93.17%, 94.60% and 92.43% for DR, MH, Optic Disc Cupping (ODC), and Within Normal Limits (WNL) classes with the augmented dataset [17]. In [18], a 20-layer CNN (termed CNN-3) was developed for feature extraction and concatenation with CNN-1. CNN-3 was also trained using the same dataset and training configurations as CNN-1.

In a study, a 25-layer custom CNN was developed to classify 32 distinct retinal diseases from the EyeNet dataset and reduce memory consumption. The model achieved a validation accuracy of 95% after 15 epochs; however, learning curves suggest that the model, along with weights and biases, was not stabilized [19]. Selecting the filter sizes, number of convolutional layers, padding, and strides is a critical task for designing a robust and memory-efficient disease identification model. To find these optimal parameters for a CNN model specifically designed for binary classification on the RFMiD, Glowworm Swarm Optimization (GSO) was implemented by [20]. The optimized model demonstrated 95.09% accuracy, although the training accuracy and loss curves indicated overfitting.

To classify 20 distinct retinal pathologies, [21] introduced a memory-efficient, low-complexity 6-layer custom CNN architecture trained on 224x224 single-channel images. Although the model achieved 92.40% accuracy, it had 49.57 million total model parameters and a size of 189.14 MB (assuming 32-bit floating point images), limiting its deployment to rich-resourced environments only. To overcome the limitations of custom CNN architectures, numerous studies were focused on transfer and ensemble learning using pre-trained deep learning architectures. In a study involving Optical Coherence Tomography (OCT) images, [22] compared Vision and Swin transformers by training both using a Generative Adversarial Network (GAN)-enhanced dataset 32x32 images to find a low-complexity model. Transfer learning was employed, with VGG16 serving as a feature extractor for classifying four retinal disease classes using OCT images. However, the model's parameter count was reduced by replacing the VGG's pre-trained classification layers with custom ones. The modified model achieved 97% accuracy but still had approximately 15.21 million parameters and an overall model size of 58.01 MB [23]. However, transfer learning yielded a parameter-efficient model, particularly in comparison to custom CNN architectures discussed earlier.

An ensemble of pre-trained DenseNet121 and EfficientNetB5 backbones (termed as DeB5-XNet) was proposed for accurate classification of multiple retinal diseases. Features extracted from both backbones were concatenated and fed to the classifier layers after dimension reduction using adaptive average and max pooling. The model demonstrated a higher accuracy score (95%) [24]. In another study on severity level detection in DR using an imbalanced dataset, [25] compared two lightweight pre-trained CNNs. MobileNet outperformed MobileNetV2, achieving 71% accuracy for multiclass classification. For binary classification of DR, [26] compared the effectiveness of pre-trained MobileNetV3 with a custom CNN. The pre-trained model yielded 97% accuracy; however, both the parameter count and the model size indicate higher computational demands.

For predicting four different retinal conditions from OCT images, [27] also employed transfer learning utilizing the VGG19 architecture. The model took 265 minutes to train on 150x150x3 images and achieved outstanding classification performance, as well as 4 minutes for inference generation. A relatively heavyweight modified VGG was introduced for multiclass retinal disease classification. The model was trained for 55 epochs on 496x496 single-channel randomly augmented images, with a batch size of 112 and the Adam optimizer. The resulting architecture had 20.29 million parameters, suggesting it is only suitable for high-end hardware [28].

To improve classification performance for normal, cataract, glaucoma, and other retinal diseases, [29] introduced DIA-VXNET, a dual-branch hybrid transfer-learning architecture that uses VGG and XceptionNET as backbones. Features were extracted from both backbones, processed via a transition block and an additional layer. Processed features were fused and fed to the classification block for generating softmax predictions. The classification block employed one flatten, three Fully Connected (FC), and two dropout layers. The DIA-VXNET achieved 99.76% accuracy on 128x128x3 input images with a batch size of 128 across a hybrid dataset, comprising images from the High Resolution Fundus (HRF), Indian Diabetic Retinopathy Image Dataset (IDRiD), and Eye Recognition datasets. Both branches extracted features from the final feature extraction layers of the backbone architectures.

For glaucoma classification, [30] fine-tuned and combined 11 pre-trained CNNs (InceptionV3, DenseNet121, DenseNet169, ResNet50, ResNet50V2, ResNet101, MobileNet, MobileNetV2, NASNetMobile, VGG16, and VGG19) using an ensemble method by using two different classification modules. The design of the first classification module was inspired by [31], while the second one included a dropout layer, as suggested by [32]. Fine-tuned NASNetMobile architectures, NASNetMobile_m1 (termed as NAS_m1) and NASNetMobile_m2 (referred as NAS_m2), achieved 91.95% and 90.23% accuracy on a hybrid dataset.

In a recent study, [33] introduced a lightweight 37-layered model for classifying DR and its associated severity levels. The custom CNN was trained using 227x227 images from the Asia Pacific Tele-Ophthalmology Society (APTOS) 2019 dataset. After training, evaluation, and optimization for DR detection, the optimized model was used to apply the learned weights to detect the severity of DR. The model achieved 80% validation accuracy but took 14.5 minutes to train. To

enable the development of CAD for DR detection, [34] introduced a hybrid model, called EffNet-SVM. The EffNet-SVM fine-tuned the pre-trained EfficientNetV2-Small. The fine-tuned model was used as a feature extractor, producing 1280 distinct features. These features were provided as input to the SVM classification model for binary classification of DR. A hybrid deep learning architecture was introduced and evaluated for binary and multi-class DR classification using three different datasets. Multiscale features were extracted, fused, and flattened after processing from the stem layers. The flattened features were fed to two Recurrent Neural Network (RNN) layers: an attention layer and a classification layer. However, the proposed framework was computationally expensive, so optimization for resource-constrained environments was also suggested [35]. In a recent study [15], a lightweight CNN comprising 1.67 million trainable parameters was proposed. A hybrid-balanced dataset (4304 images) was used for model training and classifying cataract, DR, NL, and other pathologies. The model training was performed for 100 epochs with a batch size of 32 using 130x130 images. It also achieved superior performance; however, considerable time was required for pre-processing the dataset images. For a highly unbalanced dataset, [36] developed and trained a lightweight CNN on 150x150x3 images after balancing the dataset using Synthetic Minority Oversample Technique (SMOTE). The CNN architecture employed Global Average Pooling (GAP) and dropout layers in the classification module. Despite using these layers, the model still had 16.5 million trainable parameters due to the inclusion of the attention modules. Another lightweight architecture was developed by using GAP and dropout layers in the classifier module for the detection of Retinopathy of Prematurity (RoP). This architecture integrates a pre-trained DenseNet121 and residual attention module for extracting disease-specific features. The model was extremely compact, with only 0.64/7.31 million trainable parameters. However, the model was trained for 100 epochs using an 80:20 train-evaluation split after balancing the dataset [37]. The model's performance was impressive, but its ability to generalize to an independent dataset was not evaluated.

## 3. Materials and methods

This section includes a detailed description of the research framework and LiteFeatNet's architectural specifications, including various modules designed and integrated for efficient and reliable detection of multiple retinal conditions. Fig 2 shows the entire pipeline and flow of this study. This pipeline highlights various aspects of this study, including pre-processing, image augmentation steps, component modules of the proposed LiteFeatNet architecture, and evaluation metrics.

### 3.1. Dataset description

To conduct experiments, the Retinal Fundus Multi-Disease Image Dataset (RFMiD), a publicly available dataset **(accessed on 02 July 2025)** is used. RFMiD is a multi-label dataset containing a total of 3200 high-resolution color fundus images belonging to 46 distinct retinal conditions including rare diseases [38–41]. RFMiD is inherently an imbalanced dataset, with several classes having very few samples (typically below 50 or even 10). It also contains well-represented categories such as DR, MH, and NL, with much higher sample counts (632 for DR, 523 for MH, and 669 for NL). These retinal pathologies were selected for this study, and the selection criteria for these categories were fact-driven and guided by numerous clinical, practical, statistical, and experimental factors.

From both clinical and practical perspectives, DR and MH are relatively common diseases affecting large populations worldwide. As discussed earlier [1], developing countries face numerous challenges due to limited resources, which may sometimes lead to misdiagnosis. These challenges require reliable automated CAD systems for effective and efficient patient management through accurate and timely diagnosis. MH, on the other hand, causes media opacities such as cataract, making it difficult to capture high-quality images of retinal structures. Images from the NL class label were selected to evaluate the effectiveness of our proposed classification model in handling fundus images with varying quality. This also ensures the model's ability to distinguish healthy retinas from diseased ones, thereby reducing false diagnoses of healthy patients. Sufficient sample counts for these categories are an important statistical factor that ensures stability in training

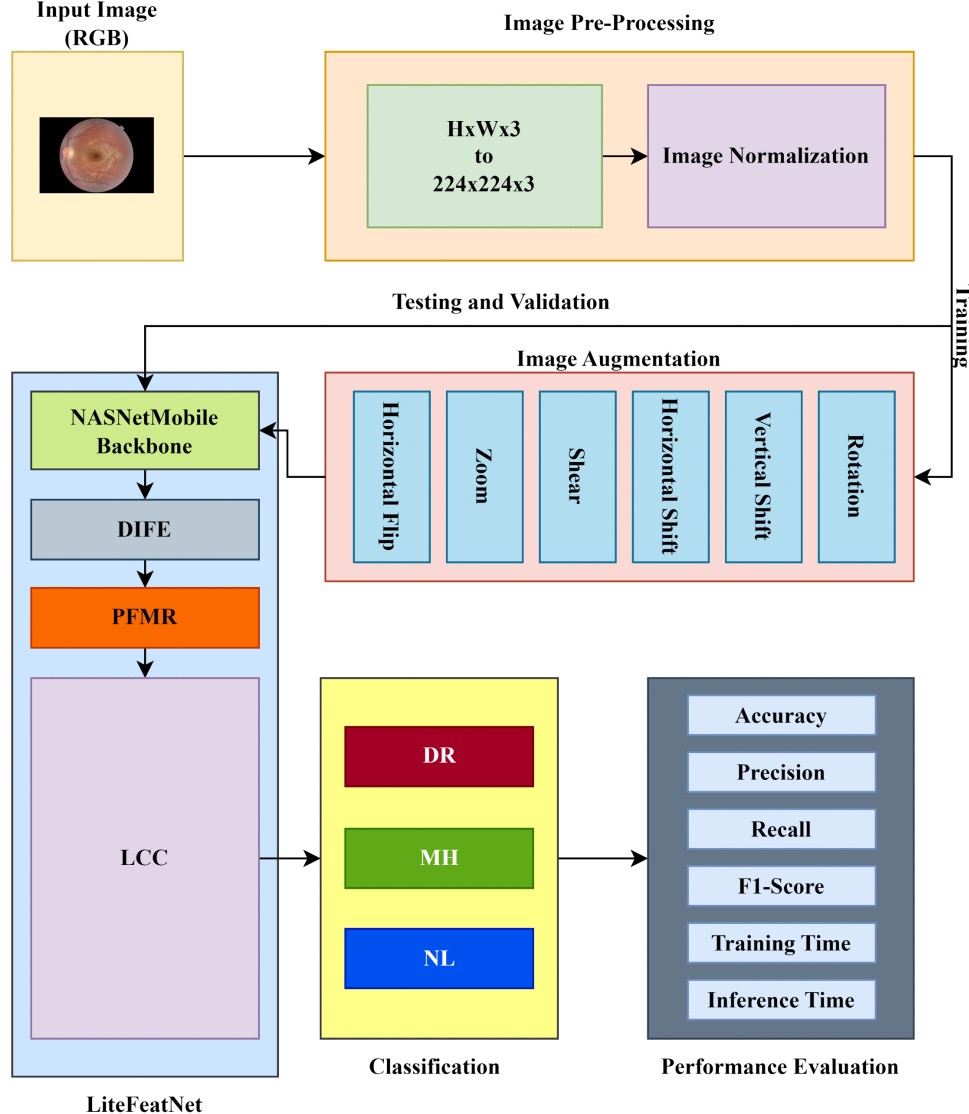

**Fig 2. Research framework and workflow pipeline.**

and reliability in evaluating the proposed LiteFeatNet model. The selection of class labels, thus performed, also enabled us to simulate a controlled experimental setting for rigorously evaluating this architecture.

The RFMiD dataset is divided into pre-defined training, validation, and testing splits by the original contributors. To evaluate our proposed fine-tuned CNN model, 1824 images for these class labels are selected directly from the predefined sub-sets, without structural amendments to the original dataset, re-splitting, or biased sample selections. This experimental protocol ensured that there was no data leakage across these splits. Out of 1824 images, the total numbers in the validation and test sets are 368 and 362, respectively, resulting in a 60:20:20 train-validation-test split. Fig 3 presents the distribution of image samples and class labels used in this study.

It is worth mentioning that both validation and test sets were non-overlapping and independent from each other. All the models used in this study were trained using 1094 training set samples, which are independent from validation and test

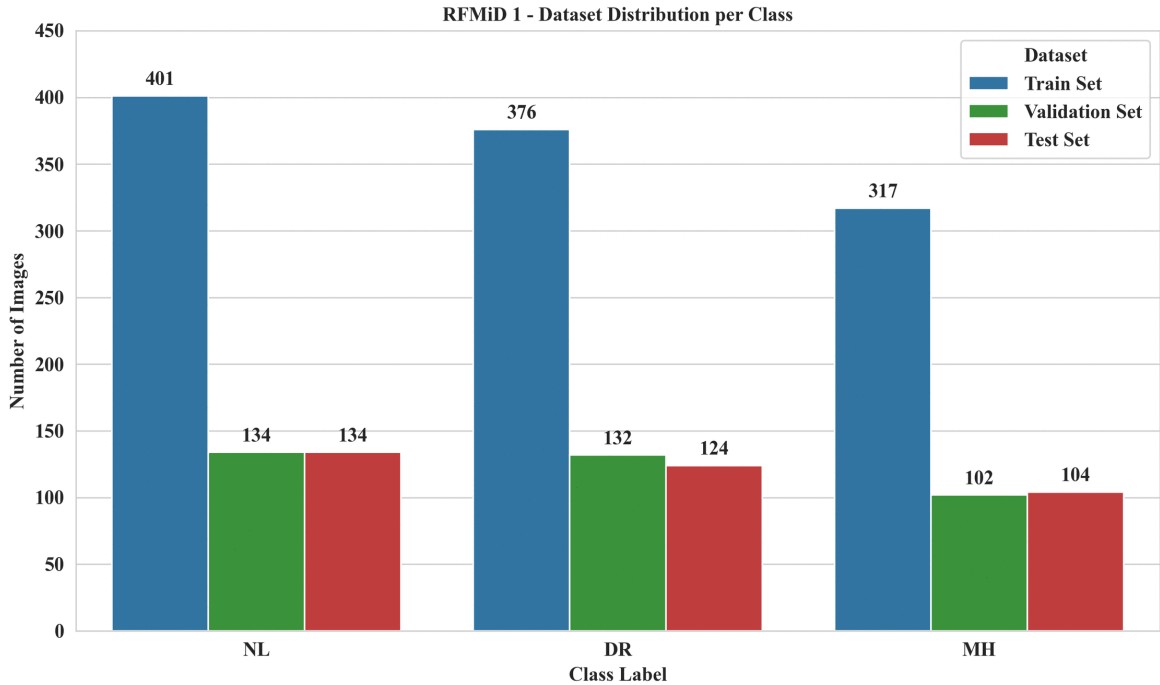

**Fig 3. Class label distribution for the study.**

sets. Validation set samples were used solely for monitoring model behaviour on unseen data during training and to find optimal hyper parameters of these models, particularly for the proposed LiteFeatNet model, whereas test samples were used to evaluate these models.

## 3.2. Dataset pre-processing

The pipeline for loading images (in RGB color space) and corresponding labels is implemented with the OpenCV (cv2), Pandas, and Numpy libraries. The color fundus images in the RFMiD are available in three different resolutions (2848x4288, 1536x2048, and 1424x2144). Additionally, the images are high-resolution, requiring high-end computing resources. To overcome this constraint and make images compatible with a deep learning model, all the images were resized to 224x224 and normalized to scale the pixel intensities, facilitating faster convergence and avoid numeric overflows/underflows during the training process. The training, validation, and testing images were converted into three corresponding Numpy arrays/Tensors. Image labels were loaded using the Pandas library, and encoded labels were also integrated into corresponding Numpy arrays/Tensors for a seamless training and evaluation loop. As a result, the compatibility of images with deep learning frameworks and optimized usage of the available memory were ensured.

## 3.3. Dataset augmentation

The aforementioned pre-processing pipeline was applied to training, validation, and testing images. Deep learning models perform well with a large number of images, so numerous augmentation strategies were applied to the training set images only to improve the model's generalizability and performance [42]. Training images were augmented on the fly in batches using the augmentation parameters given in Table 1.

**Table 1. Image augmentation parameters.**

| Augmentation Type | Parameter |
|---|---|
| Random Rotation | 0 to 30 degree |
| Horizontal Shift | 0 to 20 degree |
| Vertical Shift | 0 to 20 degree |
| Zoom Up | 0 to 20 percent |
| Random Flipping | Horizontal |

## 3.4. Deep Intermediate Feature Extraction (DIFE)

In transfer learning, multi-level features extracted using custom modules boost classification performance for multi-categorical classification of brain Magnetic Resonance Imaging (MRI) images. The incorporation of general and specific feature-extraction modules improved the model's predictive performance [43].

To develop a resource-efficient and reliable fundus disease classifier, NASNetMobile was used as the backbone feature extractor due to its proven accuracy (91.9% top-5), lower parameter count (5.3 million), and compact size (23 MB). To implement a low-learning strategy for efficient model training, pre-learned weights from the ImageNet dataset were utilized. The final feature-extraction layer of NASNetMobile generates an activated feature map of 7x7x1056 resulting from deep concatenated features. In a study, [16] extracted 7x7x1056 Rectified Linear Unit (ReLU)-activated features from the final layer of the feature extractor.

However, to make the training process time-efficient and the classifier more robust, the NASNetMobile backbone architecture was carefully analysed. Our approach extracts deep intermediate features from the backbone in a time-efficient manner without losing spatial or channel information, and fully leverages the refined features extracted by the backbone. As a result, deep intermediate features with a spatial dimension of 7x7x1056 were extracted prior to the ReLU activation from the **normal_cat_12 layer** of the NASNetMobile backbone, ensuring that the features were sufficiently deep to capture all relevant pathological patterns properly while reducing computational time. Fig 4 shows the feature extraction mechanism of the proposed DIFE.

## 3.5. Pointwise Feature Map Reduction (PFMR) module

CNNs learn discriminative features by transforming input images into different feature spaces through the stacking of multiple convolutional and pooling layers, involving dimension reduction. Therefore, implementing a low-loss dimension reduction strategy is extremely challenging [44]. Although the intermediate features from the backbone were down sampled and provided a suitable balance between feature abstraction and spatial dimensions, feeding these high-dimensional features directly to the classifier module increases the model parameters, corresponding size, and processing time. For histopathology classification, features extracted by ConvNeXt-Tiny, Vision Transformer (ViT), and Swin-Tiny were concatenated, resulting in large number of features. To find the most important, discriminative, and non-redundant features, feature reduction was implemented using Iterative Neighborhood Component Analysis (INCA). The refined features were provided as input to the ML-based classifier [45]. To achieve parameter reduction and computational efficiency, this study integrates the PFMR module for spatial-aware dimension reduction and feature refinement. The PFMR module, as shown in Fig 5, performs two operations on the input feature maps.

First, it employs a point-wise 1x1 convolutional layer that outputs refined feature maps with 512 channels, with spatial dimensions identical to those of the DIFE-extracted features. The convolutional layer refines the input features by removing redundant and low-information feature maps, thereby producing a set of feature maps with the most relevant and important discriminative features. Secondly, the output feature maps are ReLU-activated in an efficient manner. The PFMR module preserves the spatial semantic richness of backbone-extracted features while reducing the model's

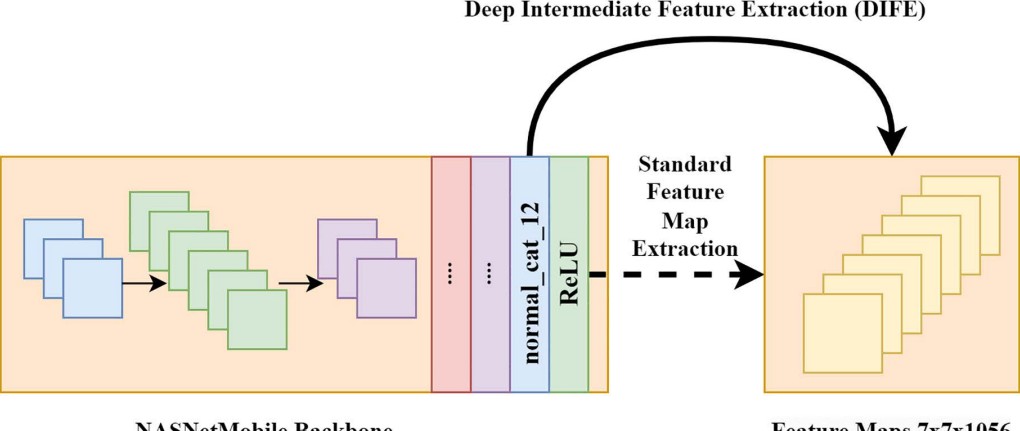

**Fig 4. DIFE from the backbone architecture.**

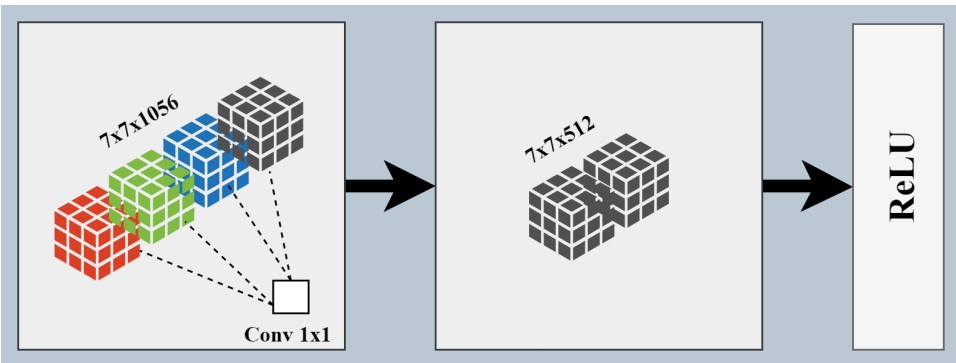

**Fig 5. PFMR module.**

parameter count, making it lighter and faster. Algorithm 1 defines the PFMR's action mechanism for dimension reduction of the input feature maps.

### Algorithm 1. PFMR Module.

```
Purpose: To reduce the channel-dimension of input features
Input: F ∈ ℝ^{7x7x1056}
1. F'=Conv₁ₓ₁(F × W) + b
            where W ∈ ℝ^{1x1x1056x512} and b ∈ ℝ^{512}
2. F''=ReLU(F')
Output: F'' ∈ ℝ^{7x7x512}
```

### 3.6. Lightweight Custom Classification (LCC) module

To reduce computational overhead and improve the model's efficiency, an LCC module is designed that processes the reduced features through a set of layers to generate predictions. Classification modules use an Artificial Neural Network (ANN) that processes input as a one dimensional feature vector. Flattening multi-channel features using the Flatten layer increases the trainable parameters exponentially. Various researchers have preferred flattening the feature maps by using a GAP layer.

In a study, [31] replaced the FC layer of pre-trained VGG16 and VGG19 with GAP layer for glaucoma classification. The GAP layer output was provided to a prediction layer with two neurons. For effective DR classification, a fine-tuned EfficientNetV2-Small was trained for 20 epochs using 224x224 augmented images (batch size: 32) from the APTOS 2019 dataset with the Adam optimizer (learning rate: 0.0001). The model included a 20% dropout, one GAP, one Batch Normalization (BN) layer, and a softmax-activated classification layer in the classification module [34].

In a study, an attention module-powered custom classifier model with a four-layer classification module having one GAP, one FC, one dropout, and a softmax layer was introduced for accurate predictions of class label [46]. In a recent study [47] the number of trainable parameters was reduced by utilizing a GAP layer with a ResNet18 backbone for binary classification of glaucoma. To accurately classify renal tumors, [48] introduced a lightweight CNN that employed four customized KidneyNeXt blocks with BN layers for feature extraction. Extracted features were processed through GAP, FC, softmax, and classification layers. The model had 7.1 million parameters and achieved superior performance (trained for 30 epochs) across multiple datasets for multi-class classification. Fig 6 shows a 6-layered classification module, and the associated mathematical details are given in Algorithm 2.

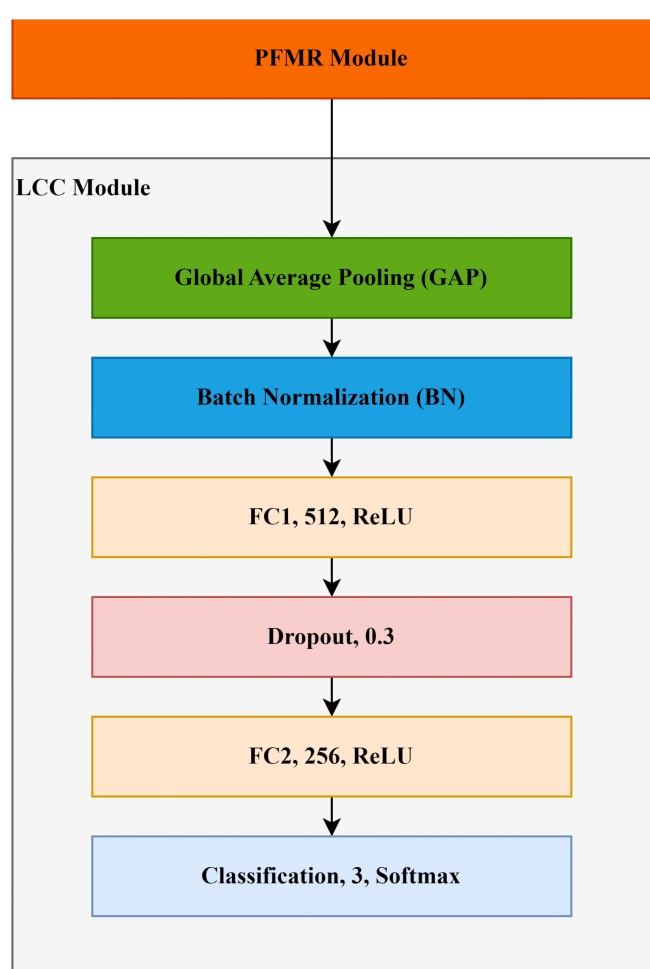

**Fig 6. Architecture of the LCC module.**

## Algorithm 2. Layer-wise breakdown of LCC module.

**Purpose:** To reduce the parameters and generate predictions
**Input:** $F'' \in \mathbb{R}^{7 \times 7 \times 512}$
**1. GAP Layer:** $z_1 = \frac{1}{7 \times 7} \sum_{i=1}^{7} \sum_{j=1}^{7} F''_{I,j,c}$
$$\text{where } z_1 \in \mathbb{R}^{512}$$
**2. BN Layer:** $z_2 = BN(z_1)$
$$\text{where } z_2 \in \mathbb{R}^{512}$$
**3. FC1 Layer:** $z_3 = ReLU(W^t z_2 + b_0)$
$$\text{where } W \in \mathbb{R}^{512 \times 512}, \ b_0 \in \mathbb{R}^{512} \text{ and } z_3 \in \mathbb{R}^{512}$$
**4. Dropout Layer:** $z_4 = Dropout(z_3, 0.3)$
$$z_4 \in \mathbb{R}^{512} \text{ where}$$
**5. FC2 Layer:** $z_5 = ReLU(W^t z_4 + b_1)$
$$\text{where } W \in \mathbb{R}^{256 \times 512}, \ b_1 \in \mathbb{R}^{256} \text{ and } z_5 \in \mathbb{R}^{256}$$
**6. Final Prediction Layer:** $\hat{y}_i = Softmax(W^t z_5 + b_2)$
$$\text{where } W \in \mathbb{R}^{3 \times 256}, \ b_2 \in \mathbb{R}^3,$$
**Output:** $\hat{y}_i$ where $\hat{y}_i \in \mathbb{R}^3$ and $\sum_{i=1}^{3} \hat{y}_i = 1$

Hence, the LCC module of the LiteFeatNet also uses a GAP layer instead of a Flatten layer to reduce model parameters further and lower the risk of overfitting, while preserving channel-wise information. The GAP layer is followed by a BN layer to stabilize the learning, control overfitting, and increase convergence to global minimum. Two FC layers with 512 and 256 neurons, with a 30% dropout layer in between were also added to ensure proper training refinement and prevent overfitting. FC layers utilized the ReLU activation function. The final classification layer, with three output nodes, was softmax-activated to generate probabilities for DR, MH and NL class labels.

### 3.7. Parameter specifications of the LiteFeatNet model

The detailed specifications of the LiteFeatNet architecture, including module-wise layer specifications with associated output shapes and the number of parameters, are given in Table 2.

### 3.8. Experimental setup

Experiments were conducted using Python 3.11.11, TensorFlow 2.18.0, and Keras 3.8.0. To train the LiteFeatNet, we used 30 epochs, a batch size of 32, Adam optimizer with an initial learning rate of 0.0001 (1e-4), and Categorical Cross-Entropy (CCE) as a loss function. Adam optimizer is highly efficient and widely used for training deep neural networks [49]. A

**Table 2. Layer-wise specification of the LiteFeatNet model.**

| Module | Layer Specification | Output Shape | Number of Parameters |
|---|---|---|---|
| DIFE from the NASNetMobile Backbone | normal_cat_12 | 7x7x1056 | 4,269,716[nt] |
| PFMR Module | Conv, 512, 1x1 | 7x7x512 | 541,184[t] |
| | ReLU | 7x7x512 | 0[t] |
| LCC module | GAP | 512 | 0[t] |
| | BN | 512 | 1,024[nt] 1,024[t] |
| | Dense, 512 | 512 | 262,656[t] |
| | Dropout, 0.3 | 512 | 0[t] |
| | Dense, 256 | 256 | 131,328[t] |
| | Dense, 3 | 3 | 771[t] |

[nt]Non-Trainable Parameters, [t]Trainable Parameters

learning rate scheduler (ReduceLROnPlateau) was also configured to adjust the learning rate based on validation accuracy, for better convergence and reducing overfitting. The ReduceLROnPlateau observes validation accuracy over three consecutive epochs to detect signs of improvement. Otherwise, it reduces the learning rate by a factor of 0.5. However, the learning rate is not allowed to fall below the minimum threshold value of 0.0000001. Mathematically, CCE can be defined as given in Eq 1, where n is the number of classes.

$$CCE(y, \hat{y}) = -\sum_{i=1}^{n} y_i \log(\hat{y}_i)$$

(1)

Further, extensive experiments are conducted with numerous architectures LightCNN [16], CNN-1 [17], DeB5-XNet [24], DIA-VXNET [29], NAS_m1 [30], NAS_m2 [30], and six well-established pre-trained models (DenseNet121, MobileNet, MobileNetV2, NASNetMobile, Swin V2-Small and Swin V2-Tiny) due to low computational overhead, particularly for medical image datasets [50].

However, for pre-trained transformer architectures, PyTorch 2.6.0 was used due to the non-availability of these architectures in Keras. Each model was trained using the same dataset splits, pre-processing pipeline, and training setup. Moreover, for pre-trained models, the Adam optimizer with a standard learning rate of 0.001 was used. Additionally, ReduceLROnPlateau and Step Decay Learning Rate schedulers were configured as learning rate schedulers for pre-trained CNNs and transformers, respectively. This rigorous experimentation pipeline was executed using the Kaggle platform, which provides an Intel Xeon Central Processing Unit (CPU), 29 Gigabytes (GB) of Random Access Memory (RAM), and NVIDIA Tesla T4 Graphics Processing Units (GPUs).

## 4. Results and discussion

The performance of the proposed integrated framework (LiteFeatNet) is evaluated using widely accepted confusion-matrix-driven classification metrics, including accuracy, precision, recall, specificity, and F1-score [51] on 362 images from the test set of the RFMiD. Additionally, the number and size of parameters are also included as metrics to demonstrate the effectiveness of strategies adopted to improve stability, reliability, and efficiency of this carefully integrated low-learning-guided lightweight retinal disease detection and identification model. Moreover, to reflect development- and deployment-level computational efficiencies, we have included training and testing/inference times, in addition to the above-mentioned performance metrics. Accuracy measures the overall predictive performance of a classifier by taking the proportion of correct predictions and total predictions made by the model. Mathematically, accuracy is measured with the help of True Positives (TP), True Negatives (TN), False Positives (FP), and False Negatives (FN), and defined by Eq 2.

$$Accuracy = \frac{TP + TN}{TP + TN + FP + FN}$$

(2)

Precision is a powerful metric for evaluating a model's reliability in terms of the number of positively predicted cases. Precision measures the proportion of actual positives (TP) among all positive predicted samples. Mathematically, precision is defined by Eq 3.

$$Precision = \frac{TP}{TP + FP}$$

(3)

Recall (also known as sensitivity) or True Positive Rate (TPR), and its counterpart, called specificity, are two robust metrics in medical image analysis. Recall quantifies the ability of a model to correctly identify the actual positive cases, whereas specificity measures the ability of a model to identify actual negative (TN) cases correctly. Sensitivity and specificity are defined mathematically in Eq 4 and Eq 5, respectively.

$$Recall \text{ or } Sensitivity = \frac{TP}{TP + FN}$$

(4)

$$\text{Specificity} = \frac{TN}{TN + FP} \tag{5}$$

F1-score measures the balance between FP and FN by measuring the harmonic mean of precision and recall. The F1-score is given by Eq 6.

$$\text{F1} - \text{Score} = 2 \text{ x } \frac{\text{Precision x Recall}}{\text{Precision + Recall}} \tag{6}$$

Before presenting the label-specific model performances directly, the results are presented in a sequence following this research approach.

## 4.1. Comparison of parameters and model size

In this section, the number and size of the LiteFeatNet's parameters (in megabytes, MB) are compared with those of various cutting-edge transfer learning and benchmark architectures to examine computational complexity and learning capability. Moreover, the number and size of the trainable, non-trainable, and total parameters across several architectures are also compared with the proposed architecture. For convenience and unit conversions, the number of parameters is also converted into millions (M). The SqueezeNet architecture included a fire module to reduce the model's parameters and associated size. Three different variants of SqueezeNet1.1 were compared by changing the number of fire modules. The SqueezeNet1.1 variant 1 (one fire module) reduced the number of trainable parameters by 54 time compared to the baseline architecture. Moreover, the size was reduced to 0.051 MB (SqueezeNet1.1 variant 1), whereas the baseline was 2.76 MB [52]. In this regard, Table 3 summarizes the comparison of the parameters and associated sizes for various architectures, including lightweight CNNs, pre-trained models, and LiteFeatNet. According to Table 3, the transformer architectures (Swin V2-Small and Swin V2-Tiny) have the highest total parameter counts (48.97M and 27.58M), indicating the highest computational cost. Architectures such as MobileNetV2 and CNN-1 [17], with total parameter counts of 2.26M and 0.49M, respectively, are among the most compact models for resource-constrained environments.

**Table 3. Comparison of number and size of parameters.**

| Model | Number of Parameters | | | Size | | |
|---|---|---|---|---|---|---|
| | Total | Trainable | Non-Trainable | Trainable Parameters | Non-Trainable Parameters | Entire Model |
| CNN-1 [17] | 499,011 | 499,011 | 0 | 1.90 | 0.00 | 1.90 |
| DenseNet121 | 7,040,579 | 3,075 | 7,037,504 | 0.01 | 26.85 | 26.86 |
| DeB5-XNet [24] | 37,140,282 | 1,582,083 | 35,558,199 | 6.04 | 135.64 | 141.68 |
| DIA-VXNET [29] | 40,923,883 | 5,344,899 | 35,578,984 | 20.39 | 135.72 | 156.11 |
| LightCNN [16] | 5,355,159 | 1,085,443 | 4,269,716 | 4.14 | 16.29 | 20.43 |
| MobileNet | 4,256,867 | 3,003 | 4,253,864 | 0.01 | 16.23 | 16.24 |
| MobileNetV2 | 2,261,827 | 3,843 | 2,257,984 | 0.01 | 8.61 | 8.63 |
| NAS_m1 [30] | 4,272,887 | 3,171 | 4,269,716 | 0.01 | 16.29 | 16.30 |
| NAS_m2 [30] | 17,549,719 | 13,280,003 | 4,269,716 | 50.66 | 16.29 | 66.95 |
| NASNetMobile | 4,272,887 | 3,171 | 4,269,716 | 0.01 | 16.29 | 16.30 |
| Proposed (LiteFeatNet) | **5,207,703** | **936,963** | **4,270,740** | **3.57** | **16.29** | **19.87** |
| Swin V2-Small | 48,970,749 | 2,307 | 48,968,442 | 0.01 | 186.76 | 186.77 |
| Swin V2-Tiny | 27,584,877 | 2,307 | 27,582,570 | 0.01 | 105.22 | 105.23 |

Parameters (in numbers), size (in MBs).

The LiteFeatNet architecture, with 5.21M total parameters, lies between lightweight transfer learning models like NASNet-Mobile, which has a total parameter count of 4.27M. Transfer learning architectures like MobileNet, MobileNetV2, NASNetMobile, and DenseNet121 have extremely low trainable parameter counts (not above 0.004M), due to fine-tuning of the classification head only. These architectures are computationally efficient and low-learning, but sometimes these models underperform, particularly when applied to other domains like medical imaging due to cross-domain adaptation issues.

The CNN-1 [17] architecture is 100%, while LightCNN [16] has 20.27% (1.08M/5.35M) trainable parameters. Similarly, LiteFeatNet's trainable parameter count approximately 17.99% of the total parameters. Moreover, in comparison to heavyweight ensemble architectures like DeB5-XNet [24] (1.58M/37.14M), DIA-VXNET [29] (5.34M/40.92M), and NAS_m2 [30] (13.28M/17.55M), the proposed LiteFeatNet model has a better trainable-to-total parameter ratio (0.94M/5.21M), making it more lightweight and parameter-efficient. This balance of trainable parameters allows excellent adaptation of the backbone to the retinal image analysis domain while remaining computationally efficient and performance-centric.

As shown in Table 3, Swin Transformer V2-Small and DIA-VXNET [29] are heavyweight models among the studied models, with the highest parameter count exceeding 37 million and sizes of 186.77 MB and 156.11 MB, respectively. The proposed LiteFeatNet model integrates a 6-layered classification block-like architectures like LightCNN [16], CNN-1 [17], DeB5-XNet [24], DIA-VXNET [29], and NAS_m2 [30], which is parameter-efficient and lightweight. So, the LiteFeatNet model is significantly lightweight (19.87 MB), particularly in comparison with DenseNet121, Swin V2-Small, Swin V2-Tiny, LightCNN [16], DeB5-XNet [24], DIA-VXNET [29], and NAS_m2 [30]. As shown in Table 3, the size of trainable parameters for the majority of pre-trained architectures are significantly low (not above 0.01 MB) in comparison to LightCNN [16], CNN-1 [17], DeB5-XNet [24], DIA-VXNET [29], NAS_m2 [30] and the LiteFeatNet architecture due to unfreezing the weights of the last layer for making these models adapt to the classification of retinal diseases. NAS_m2 [30] had 50.66 MB of trainable parameters, making it computationally intensive compared to the other models included in this study.

Although the LiteFeatNet architecture is slightly heavier than MobileNet, MobileNetV2, and CNN-1 [17], it has achieved the best performance parameters for classifying DR, MH and NL. Although CNN-1 [17] had a considerably lower size of trainable parameters in comparison to the LiteFeatNet (1.90 MB vs. 3.57 MB), it could not learn the label-specific features and resulted in lower accuracy in comparison to the LiteFeatNet's accuracy and other architectures, particularly LightCNN [16], DeB5-XNet [24], and DIA-VXNET [29]. This, again, proves its robustness and compatibility with resource-scarce computing environments. The specifications of various resource-scarce target deployment platforms for the proposed LiteFeatNet model, along with resource consumption reported by numerous studies, are given in Table 4.

The total memory $M_t$ occupied by a model during inference is given by Eq (7), where $M_p$ and $M_a$ are the model's parameters and activation memories, respectively, and $M_o$ denotes the memory overhead. In the worst-case scenario (1024x1024 image size), $M_a$ approximately reaches $\alpha = 16$ times the $M_p$ [58], defined by Eq (8).

$$M_t \approx M_p + M_a + M_o \tag{7}$$

$$M_a \approx \alpha \times M_p \tag{8}$$

Our proposed model ($M_p = 19.87$ MB) is estimated to require $M_t = 337.79 + \lambda$ MB ($M_o = \lambda$) of RAM during inference with the worst-case scenario. However, for the standard 224x224 image size, $M_a < M_p$ [58], and the inference memory for this size is always estimated to be below that for the worst-case scenario. Thus, the memory consumption threshold limits of our model are given by Eq (9), where $M_o = \lambda$ and $M_a = \beta$. Moreover, compared with the device specifications and memory consumptions reported in Table 4, our architecture is suitable for operating on the aforementioned resource-scarce computing devices, across image resolutions ranging from 224x224 to 1024x1024.

$$(19.87 + \beta + \lambda) \lesssim M_t \lesssim (337.79 + \lambda) \tag{9}$$

**Table 4. Target deployment platforms.**

| Research study | Device/ platform | Device specifications | Use case | Resource consumption reported |
|---|---|---|---|---|
| [53] | Raspberry Pi 4 Model B | Cortex A72 1.5 GHz (quad core), 4GB RAM (3200 MHz), No GPU | Image classification using CNN | 512 MB of RAM |
| [53] | NVIDIA Jetson Nano | Cortex A57 1.43 GHz (quad core), 4GB RAM (1600 MHz), and Maxwell GPU with 128 cuda cores | Image classification using CNN | 700 MB of RAM |
| [54] | Raspberry Pi 4 | Cortex A72 1.5 GHz (quad core), 4GB RAM (3200 MHz), No GPU | Image classification using CNN | 2.5 GB of RAM |
| [54] | NVIDIA Jetson Nano | Cortex A57 1.43 GHz (quad core), 4GB RAM (1600 MHz), and Maxwell GPU with 128 cuda cores | Image classification using CNN | 3.5 GB of RAM |
| [55] | Xiaomi Black Shark | Qualcomm Snapdragon 845 octa-core 2.8GHz, 8GB RAM (1866 MHz), Qualcomm Adreno 630 GPU, 256 GB Storage | Object detection using YOLO v3 | 633 MB of RAM with OpenCV 263 MB of RAM with TensorFlow lite |
| [56] | Raspberry Pi 5 Model B Rev 1.0 | Broadcom Cortex A76 2.4 GHz (quad-core), 8 GB RAM, VideoCore VII GPU, 64 GB Storage | Image classification using CNN | 1948 MB of RAM |
| [56] | NVIDIA Jetson Nano 4.6.6 | Cortex A57 (MP core), 4 GB RAM (1600 MHz), and Maxwell GPU with 128 cuda cores, 64 GB Storage | Image classification using CNN | 1960 MB of RAM |
| [57] | Raspberry Pi 4 Model B | Cortex A72 1.5 GHz (quad core), 4GB RAM (3200 MHz), No GPU | Image classification using CNN | – |

Hence, the parameter trade-off achieved with the PFMR and LCC modules reduced the number and size of parameters sufficiently, making it suitable for memory-efficient and faster computations on resource-constrained across numerous computing devices, while remaining complex enough to learn inherent label-specific patterns. Moreover, the performance evaluation metrics (as described in the coming sections) indicate that the LiteFeatNet architecture is compact, sufficiently trainable, and well-adapted to the classification of multiple retinal conditions.

## 4.2. Training performance

The trend analysis of training and validation accuracies for all the models, including LiteFeatNet are illustrated in Fig 7. According to Fig 7K, the LiteFeatNet model achieved over 80% training and validation accuracy within the first eight epochs, suggesting faster convergence.

Hence, the proposed architecture was able to learn the discriminative features earlier during the training process. Despite minor fluctuations, the gap between the two accuracies narrows and stabilizes after epoch 10, and the trend shows continuous improvement. Both accuracy curves are well-aligned and are not significantly apart, intersecting at multiple points (particularly, epoch 20 and epoch 30), reflecting well-generalized training behavior and better convergence with no signs of overfitting.

Moreover, both accuracy curves plateau between 85–87% during later epochs, assuring the stability and robustness of the proposed framework. Fig 8 presents the trajectories of the training and validation losses for several models evaluated in this study.

According to Fig 8K for our integrated framework (LiteFeatNet), both losses exhibit a steep decline during the first five epochs, with the training and validation losses dropping from 0.82 and 0.74, respectively. This behavior of the loss curves reflects that this model has effectively adjusted to minimize prediction errors. Till epoch 17, both training and validation losses continue to decrease gradually, exhibiting a consistent downward trend with some fluctuations. From epochs 17–20, both loss curves remain aligned and show no signs of significant divergence. After epoch 20, both the training and validation losses plateau at around 0.25–0.30, suggesting good convergence and generalization.

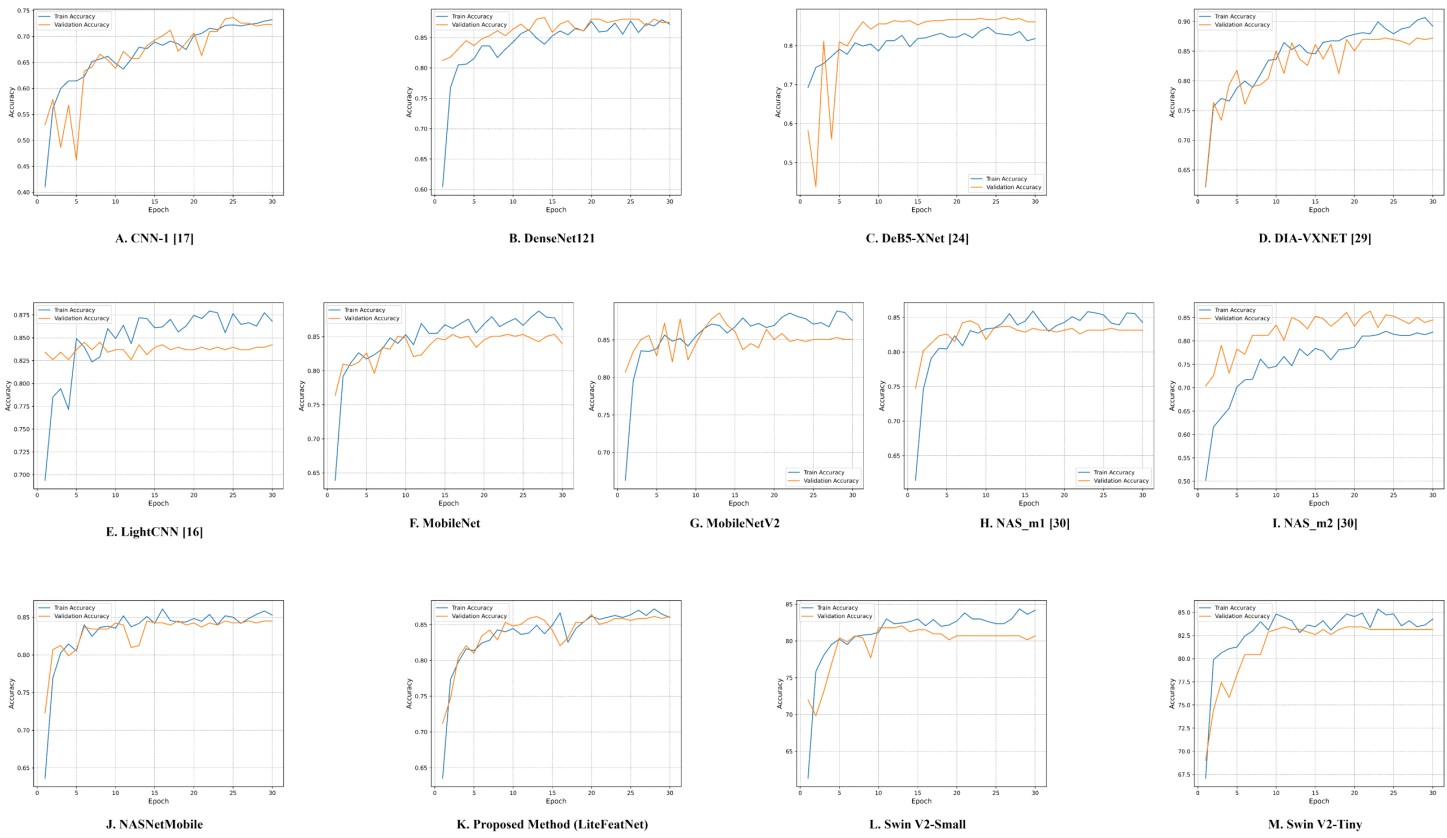

**Fig 7. Accuracy plots.** (A) CNN-1 [17] (B) DenseNet121 (C) DeB5-XNet [24] (D) DIA-VXNET [29] (E) LightCNN [16] (F) MobileNet (G) MobileNetV2 (H) NAS_m1 [30] (I) NAS_m2 [30] (J) NASNetMobile (K) Proposed Model (LiteFeatNet) (L) Swin V2-Small (M) Swin V2-Tiny.

Both training curves, as presented in Fig 7K and Fig 8K, suggest that our LiteFeatNet model has a better learning capability and avoids overfitting. This pattern confirms the high effectiveness of the LiteFeatNet model in generalizing to unseen data, thanks to its robust feature-extraction and classification layers. As accuracy and loss curves do not indicate inconsistent behavior or abrupt drops in validation performance, this further reinforces its reliability.

According to Fig 7, all the architectures also demonstrated a stable learning behavior in terms of accuracy. Moreover, both training and validation accuracies for all evaluated models reach performance plateaus, particularly in later epochs (20 or above). Similarly, the loss curves in Fig 8, provide further confirmation of stable optimization and convergence of all the evaluated architectures. Therefore, both accuracy and loss plots clearly indicated convergence across all the models.

### 4.3. Confusion matrix

A confusion matrix provides a comparative label-wise summary of the classifier's performance across different class labels. The confusion matrix for test data predictions corresponding to three class labels is given in Fig 9.

According to Fig 9, the LiteFeatNet was successful in correctly classifying 102 out of 124 DR, 95 out of 104 MH, and 130 out of 134 NL class instances. These stats reflect the model's high classification accuracy and minimal confusion among the class labels. Notably, the majority of misclassifications occurred between the DR and MH class labels, likely due to overlapping disease patterns across test images. On the other hand, the model misclassified only 4 NL images, indicating the model's superior ability to characterize healthy images.

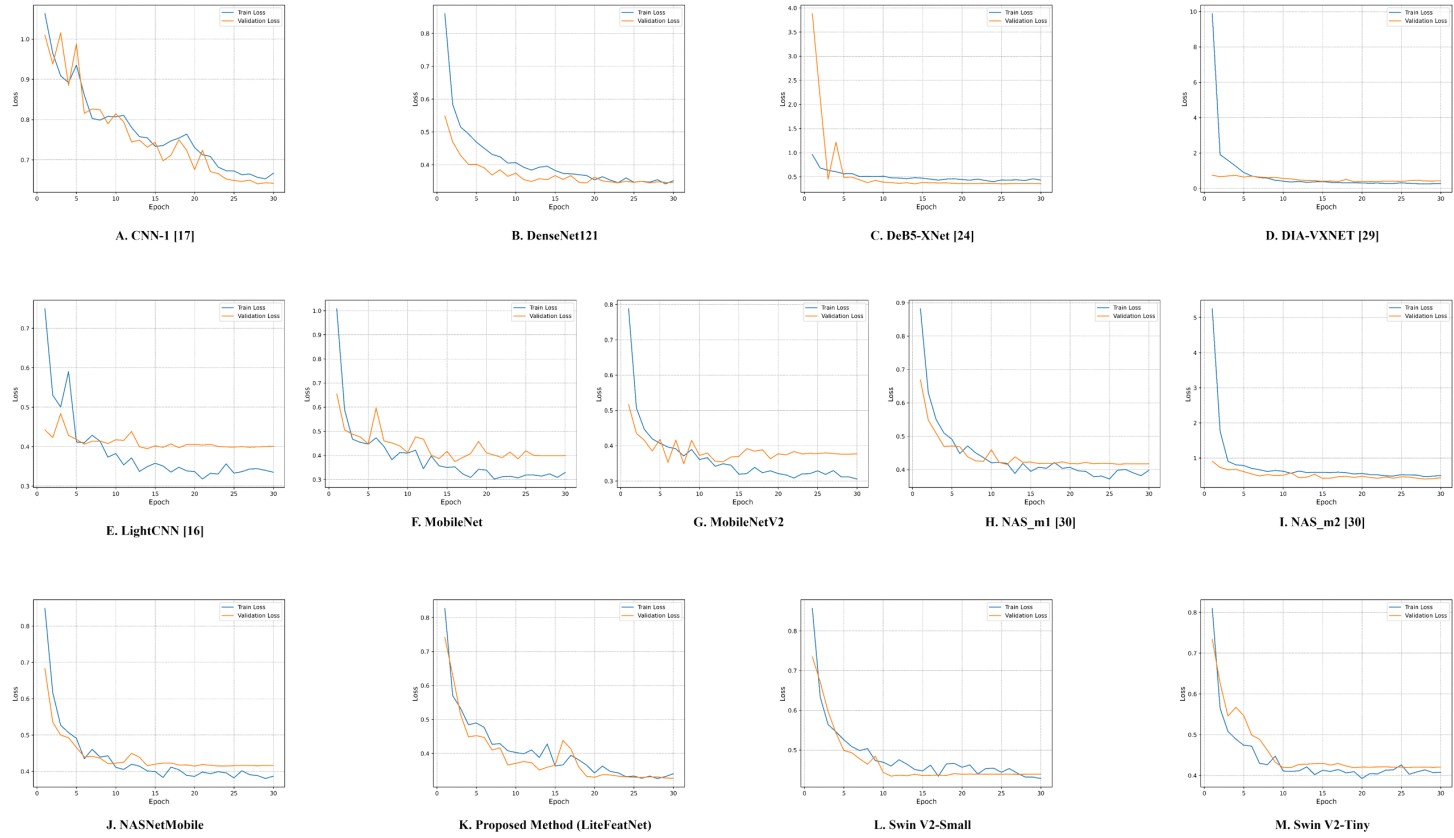

**Fig 8. Loss plots.** (A) CNN-1 [17] (B) DenseNet121 (C) DeB5-XNet [24] (D) DIA-VXNET [29] (E) LightCNN [16] (F) MobileNet (G) MobileNetV2 (H) NAS_m1 [30] (I) NAS_m2 [30] (J) NASNetMobile (K) Proposed Model (LiteFeatNet) (L) Swin V2-Small (M) Swin V2-Tiny.

## 4.4. Overall performance metrics

In this section, the performance metrics selected for the study are presented to provide detailed insights, as shown in Table 5. According to Table 5, lightweight transfer learning architectures, such as NASNetMobile, MobileNet, and Mobile-NetV2, performed faster and achieved higher accuracies than transformer architectures.

The LightCNN [16] achieved better performance metrics (accuracy: 87.57%, precision: 87.55%, and F1-Score: 87.52%), exhibiting better classification performance, compared to numerous pre-trained CNN and transformer architectures.

The proposed LiteFeatNet model demonstrated the highest classification performance (accuracy: 90.33%, precision: 90.69%, and F1-score: 90.27%), outperforming various pre-trained architectures and the LightCNN [16]. It demonstrated exceptional efficiency and effectiveness with only (0.94M/5.21M) trainable-to-total parameters and a compact model size of 19.87 MB. LiteFeatNet was also evaluated against an existing architecture, CNN-1 [17]. Although the CNN-1's [17] processing time was significantly lower, it failed to generalize to the test dataset, resulting in poor performance. LiteFeatNet significantly improved the accuracy of CNN-1 [17] by 26.74% and precision by 25.24%. Similarly, the F1-score of the proposed LiteFeatNet reflects that precision and recall scores are well-balanced. Additionally, our model also outperformed two transfer learning driven-hybrid ensemble architectures, DeB5-XNet [24] and DIA-VXNET [29], in all respects, including parametric and performance metrics. The proposed model exhibited excellent performance, thereby surpassing accuracies (DeB5-XNet [24]: 88.67% and DIA-VXNET [29]: 88.40%) and precision (DeB5-XNet [24]: 89.19% and DIA-VXNET

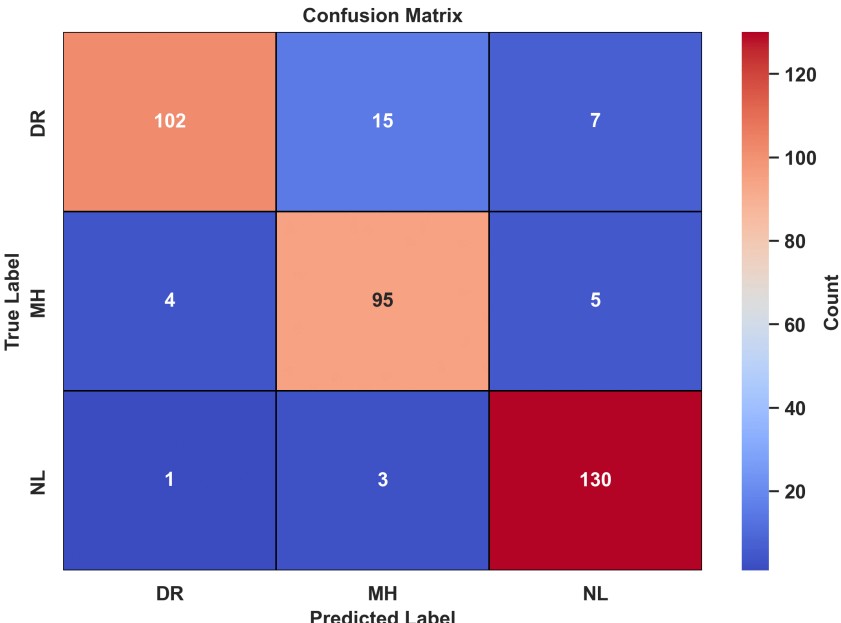

**Fig 9. Confusion matrix for LiteFeatNet using RFMiD.**

**Table 5. Overall performance metrics for RFMiD.**

| Model | Accuracy | Precision | Recall | F1-Score | Training Time | Inference Time |
|---|---|---|---|---|---|---|
| CNN-1 [17] | 71.27 | 72.41 | 71.27 | 70.89 | 325.65 | 0.91 |
| DenseNet121 | 89.23 | 89.69 | 89.23 | 89.11 | 416.62 | 1.59 |
| DeB5-XNet [24] | 88.67 | 89.19 | 88.67 | 88.61 | 635.35 | 2.96 |
| DIA-VXNET [29] | 88.40 | 88.80 | 88.40 | 88.30 | 553.30 | 4.21 |
| LightCNN [16] | 87.57 | 87.55 | 87.57 | 87.52 | 461.11 | 1.32 |
| MobileNet | 84.81 | 85.60 | 84.81 | 84.50 | 350.27 | 1.00 |
| MobileNetV2 | 85.08 | 85.59 | 85.08 | 84.88 | 385.75 | 1.05 |
| NAS_m1 [30] | 86.46 | 86.47 | 86.46 | 86.45 | 399.68 | 1.30 |
| NAS_m2 [30] | 86.46 | 86.53 | 86.46 | 86.44 | 410.72 | 1.31 |
| NASNetMobile | 86.74 | 86.70 | 86.74 | 86.68 | 406.42 | 1.34 |
| Proposed (LiteFeatNet) | **90.33** | **90.69** | **90.33** | **90.27** | **409.35** | **1.32** |
| Swin V2-Small | 82.04 | 85.50 | 82.04 | 81.67 | 498.06 | 3.35 |
| Swin V2-Tiny | 82.04 | 83.74 | 82.04 | 81.83 | 252.16 | 2.15 |

Accuracy, Precision, Recall, and F1-Score (**%**), Training Time, and Inference Time (seconds)

[29]: 88.80%). Both ensemble architectures' performance was excellent and close to the proposed model, but both architectures had significantly larger trainable parameter sizes (3.57 MB vs. 6.04 MB and 20.39 MB).

The performance of NAS_m1 [30] and NAS_m2 [30] architectures was quite close to that of the pre-trained NASNetMobile. NAS_m1 [30] exhibited slightly lower precision as compared to NAS_m2 (86.47% vs. 86.53%) despite its lower size; however, NASNetMobile outperformed these two architectures.

Swin Transformer V2-Small also showed improved precision (85.50% vs. 83.74%) as compared to the Swin Transformer V2-Tiny version; however, the accuracy of both architectures was the same (82.04%). Despite being powerful, these models were not found suitable for production and deployment in resource-constrained environments due to their larger sizes (105.23 MB and 186.77 MB, respectively).

However, the LiteFeatNet architecture demonstrated superior performance. It also outperformed all the baseline architectures, achieving the highest and optimal values for performance evaluation metrics, with a compact total size of 19.87 MB, clearly highlighting its compactness, robustness, and reliability for retinal disease classification.

According to Table 5, the computational times of NASNetMobile (inference time: 1.34 seconds and training time: 406.42 seconds), NAS_m1 (inference time: 1.30 seconds and training time: 399.68 seconds) [30], NAS_m2 (inference time: 1.31 seconds and training time: 410.72 seconds) [30], and LiteFeatNet (inference time: 1.32 seconds and training time: 409.35 seconds) were quite close. The proposed integrated architectural framework outperformed these models, with an impressive inference time, which is modestly higher, particularly compared to NAS_m1 [30] and NAS_m2 [30]. MobileNet had the lowest testing time (1.00 seconds), while Swin V2-Small had the longest inference and training times (3.35 and 498.06 seconds, respectively) among all the pre-trained architectures studied.

Both ensemble architectures, DeB5-XNet [24] and DIA-VXNET [29], required extended inference time (2.96 and 4.21 seconds, respectively), with DeB5-XNet [24] and DIA-VXNET [29] exhibiting the slowest training time (635.35 seconds) and inference time (4.21 seconds), respectively, among the customized architectures.

Although the inference and training times for our optimized learning pipeline were considerably higher than those of CNN-1 [17] (1.32 vs. 0.91 seconds) and (409.35 vs. 325.65 seconds), the LiteFeatNet model achieved far better performance metrics with competitive computational efficiency. Moreover, LiteFeatNet, despite having fewer parameters, is substantially more efficient than both DeB5-XNet [24] and DIA-VXNET [29], offering much faster inference speed (1.32 vs. 2.96 seconds) when compared with DeB5-XNet [24]. The performance metrics suggest that LiteFeatNet is not only lightweight but also efficient and more reliable for retinal disease classification.

Interestingly, the DenseNet121 has achieved extremely competitive performance scores (accuracy: 89.23%, precision: 89.69%, recall: 89.23%, and F1-score: 89.11%). Still, our proposed modular framework demonstrates improved predictive capabilities, achieving slightly higher scores for performance metrics (accuracy: 90.33%, precision: 90.69%, recall: 90.33%, and F1-score: 90.27%). In addition to this performance gain (accuracy: 1.23%, precision: 1.11%, recall: 1.23%, and F1-score: 1.30%), the proposed LiteFeatNet also exhibits deployment and development computational efficiencies, with a lower inference time (1.32 vs. 1.59 seconds) and fast-paced training (409.35 vs. 416.62 seconds). Furthermore, our deep learning-based decision-support framework also offers 30% (approximately) reduced memory footprint compared to DenseNet121. This modest size reduction is extremely important for deploying in real-world clinical settings. Apart from compact storage requirements, small models require less time to load and generate faster inferences, making them more suitable for integration with resource-scarce clinical settings, mass retinal scanning programs, and other situations requiring swift operations. Such practical scenarios rely on portable diagnostic tools, low-resourced CAD systems, and edge devices. In such real-time scenarios, models having fewer parameters, limited data movements, and cache optimization are usually preferred choice for integration because of their minimal impact on the device responsiveness [59]. With ongoing research on lightweight models, models with parameter counts ranging from 0.91 million to 71.3 million are reported to have real-time applications [60]. LiteFeatNet's parameter count (5.2 million) and corresponding $M_p$ effectively fall within this range, making $M_t$ compatible with the deployment platforms given in Table 4.

The results verify that the LiteFeatNet possesses superior capability to learn disease-wise discriminative features. Moreover, incorporating DIFE from the backbone, an optimized feature refinement strategy (PFMR), and an LCC module enabled the model to achieve robust performance for classifying NL, DR, and MH, with reduced computational cost,

particularly the inference time (1.32 seconds), making it a suitable choice for deployment in resource-scarce environments, particularly with a higher patient counts.

## 4.5. Class-wise performance metrics

Table 6 presents a comparative analysis of the performance of various deep learning models using class-wise performance metrics: accuracy, precision, sensitivity, and specificity for the class labels: DR, MH, and NL.

According to Table 6, the NASNetMobile demonstrated balanced performance, achieving higher and stable accuracy, sensitivity, and specificity for DR (89.23%, 82.26%, and 92.86%), MH (91.71%, 84.61%, and 94.57%), and NL (92.54%) classes. It suggests that NASNetMobile was robust in detecting TPs and FPs. The accuracy, precision, sensitivity, and specificity for both NAS_m1: (DR: 88.95%, 83.33%, 84.68%, and 91.18%), (MH: 91.71%, 87.00%, 83.65%, and 94.96%), and (NL: 92.26%, 88.97%, 90.30%, and 93.42%) [30] and NAS_m2: (DR: 89.50%, 83.59%, 86.29%, and 91.18%), (MH: 91.71%, 88.54%, 81.73%, and 95.74%), and (NL: 91.71%, 87.68%, 90.30%, and 92.54%) [30] also reported a balanced and similar performance metrics, however, NAS_m2 [30] reported better accuracy and precision for both DR and MH class labels. Despite sharing the same backbone, the proposed model achieved higher accuracy (DR and MH: 92.54% and NL: 95.58%) than all these models.

The MobileNet architecture demonstrated better discriminative ability in distinguishing diseased from non-diseased samples. It exhibited excellent performance, achieving a high sensitivity (97.01%) for the NL class. It also demonstrated high precision (90.72%) and specificity (96.22%) for the DR class. Similarly, MobileNetV2 demonstrated competitive performance and achieved a sensitivity of 97.01% for the NL class label. Moreover, it also performed better at classifying both DR and NL class labels, achieving a higher accuracy (87.84% and 91.16%, respectively) than the MobileNet architecture (87.57% and 90.05%, respectively).

It also reported an excellent specificity of 97.29% for the MH class label, indicating its ability to lower FP predictions. The DenseNet121 showed reliable performance, achieving higher accuracies for all three class labels (DR: 91.71%; MH and NL: 93.37%) but at the cost of higher memory consumption and computational time. Moreover, it achieved higher specificity scores for these class labels, along with higher sensitivity (90.38% and 97.01%) for MH and NL. It also achieved higher specificity for DR (97.90%) and MH (94.57%) compared to NL (91.23%). The proposed CNN-based model demonstrated higher specificity for DR (97.90%) and NL (94.74%) than DenseNet121. Although its specificity for MH was lower than DenseNet121, but LiteFeatNet's computational time was significantly lower (1.32 vs. 1.59 seconds).

Swin V2-Small exhibited excellent performance (accuracy of 91.16% and specificity of 94.96%) for the MH class. It also showed higher specificity (99.58%) and precision (98.75%) for the DR, but its sensitivity was much lower (63.71%), suggesting it was unable to classify positive samples for the DR class correctly. Swin V2-Tiny also shows lower sensitivity (74.19% and 72.11%) for the DR and MH classes, indicating higher FN predictions for both labels. Moreover, the precision (75.14%) for the NL class label was considerably low, suggesting a 25% (approx.) probability of the model's misclassification of diseased samples as NL.

The LightCNN [16] architecture achieved a balanced performance for all classes. It achieved higher performance metrics: accuracy (92.82%) and sensitivity (92.54%) for the NL class than for the DR (90.05% and 83.06%) and MH (92.26% and 86.54%) classes, as shown in Table 6. The proposed model demonstrated higher accuracy and sensitivity for both MH (92.54% and 91.35%) and NL (95.58% and 97.01%) class labels, clearly outperforming the LightCNN [16], with faster training time (409.35 seconds vs. 461.11 seconds).

Compared with the LiteFeatNet, CNN-1 [17] reported poor performance, particularly for the NL class, achieving an accuracy of 78.45%, precision of 66.09% and specificity of 74.12%. Moreover, it also exhibited lower sensitivity scores (58.87% and 67.31%) for DR and MH classes. The performance indicates that the model has poor generalization and limited capability to interpret complex disease patterns.

**Table 6. Class-wise performance metrics for RFMiD.**

| Model | Class | Accuracy | Precision | Sensitivity | Specificity |
|---|---|---|---|---|---|
| CNN-1 [17] | DR | 79.83 | 76.84 | 58.87 | 90.76 |
| | MH | 84.25 | 75.27 | 67.31 | 91.08 |
| | NL | 78.45 | 66.09 | 85.82 | 74.12 |
| DenseNet121 | DR | 91.71 | 95.19 | 79.84 | 97.90 |
| | MH | 93.37 | 87.04 | 90.38 | 94.57 |
| | NL | 93.37 | 86.67 | 97.01 | 91.23 |
| DeB5-XNet [24] | DR | 91.99 | 93.58 | 82.26 | 97.06 |
| | MH | 93.37 | 90.82 | 85.58 | 96.51 |
| | NL | 91.99 | 83.87 | 97.01 | 89.03 |
| DIA-VXNET [29] | DR | 91.71 | 93.52 | 81.45 | 97.06 |
| | MH | 92.54 | 88.12 | 85.58 | 95.35 |
| | NL | 92.54 | 84.97 | 97.01 | 89.91 |
| LightCNN [16] | DR | 90.05 | 87.29 | 83.06 | 93.70 |
| | MH | 92.26 | 86.54 | 86.54 | 94.57 |
| | NL | 92.82 | 88.57 | 92.54 | 92.98 |
| MobileNet | DR | 87.57 | 90.72 | 70.97 | 96.22 |
| | MH | 91.99 | 86.41 | 85.58 | 94.57 |
| | NL | 90.05 | 80.25 | 97.01 | 85.96 |
| MobileNetV2 | DR | 87.84 | 83.90 | 79.84 | 92.02 |
| | MH | 91.16 | 91.86 | 75.96 | 97.29 |
| | NL | 91.16 | 82.28 | 97.01 | 87.72 |
| NAS_m1 [30] | DR | 88.95 | 83.33 | 84.68 | 91.18 |
| | MH | 91.71 | 87.00 | 83.65 | 94.96 |
| | NL | 92.26 | 88.97 | 90.30 | 93.42 |
| NAS_m2 [30] | DR | 89.50 | 83.59 | 86.29 | 91.18 |
| | MH | 91.71 | 88.54 | 81.73 | 95.74 |
| | NL | 91.71 | 87.68 | 90.30 | 92.54 |
| NASNetMobile | DR | 89.23 | 85.71 | 82.26 | 92.86 |
| | MH | 91.71 | 86.27 | 84.61 | 94.57 |
| | NL | 92.54 | 87.94 | 92.54 | 92.54 |
| Proposed (LiteFeatNet) | DR | **92.54** | **95.33** | **82.26** | **97.90** |
| | MH | **92.54** | **84.07** | **91.35** | **93.02** |
| | NL | **95.58** | **91.55** | **97.01** | **94.74** |
| Swin V2-Small | DR | 87.29 | 98.75 | 63.71 | 99.58 |
| | MH | 91.16 | 86.73 | 81.73 | 94.96 |
| | NL | 85.63 | 72.28 | 99.25 | 77.63 |
| Swin V2-Tiny | DR | 86.19 | 83.64 | 74.19 | 92.44 |
| | MH | 90.88 | 94.94 | 72.11 | 98.45 |
| | NL | 87.02 | 75.14 | 97.01 | 81.14 |

Accuracy, Precision, Sensitivity, and Specificity (%).

Ensemble architectures DeB5-XNet [24] and DIA-VXNET [29] demonstrated competitive, more balanced performance in comparison to pre-trained models. The DeB5-XNet [24] achieved considerably higher performance for DR (accuracy: 91.99% and specificity: 97.06%) and MH (accuracy: 93.37% and specificity: 96.51%) with a testing time of 2.96 seconds. However, for the NL class label, this model exhibited comparatively lower precision (83.87%) and specificity (89.03%). Similarly, it also reported slightly lower sensitivities of 82.26% and 85.58% for the DR and MH class labels, respectively. The DIA-VXNET [29] also achieved a considerably lower sensitivity for DR (81.45%) and MH (85.58%). Additionally, it exhibited lower precision 84.97% and specificity 89.91% for the NL class compared to the proposed architecture (precision: 91.55% and specificity: 94.74%). However, both DeB5-XNet [24] and DIA-VXNET [29] required extensive training sessions (635.35 and 553.30 seconds) and longer testing times (2.96 and 4.21 seconds). LiteFeatNet achieved the highest accuracy (95.58%), sensitivity (97.01%), and specificity (94.74%) for the NL class label among all models, thereby demonstrating its ability to accurately predict and distinguish between healthy and diseased images. In addition to the NL class label, the LiteFeatNet successfully maintained its performance on the DR and MH classes, achieving 92.54% accuracy for both classes.

The LiteFeatNet not only matched the sensitivity (82.26%) of DeB5-XNet [24] for DR, but also exhibited better precision (95.33% vs. 93.58%) and specificity (97.90% vs. 97.06%). Similarly, compared with DIA-VXNET [29], LiteFeatNet showed improved sensitivity (91.35% vs 85.58%) for the identification of MH class label. Although the proposed model achieved lower precision (84.07%) for MH compared to DenseNet121 (87.04%), Swin V2-Small (86.73%), Swin V2-Tiny (94.94%), DIA-VXNET (88.12%) [29], and DeB5-XNet (90.82%) [24], it was much more time efficient. Hence, even with a small number of trainable parameters, the LiteFeatNet is extremely memory-efficient, reliable, and exhibits robust performance in the classification of DR, MH, and NL.

## 5. Generalizability study

To assess the generalizability and robustness of the LiteFeatNet, numerous experiments were also conducted using an external dataset. This assessment was crucial for benchmarking LiteFeatNet's adaptability and evaluating the performance and effectiveness of the proposed framework in real-world scenarios. Evaluation metrics were computed for our model using the external dataset and compared with various architectures, as discussed previously.

### 5.1. Dataset description

To perform this assessment, another publicly available dataset, RFMiD 2.0 **(accessed on 02 July 2025)** was utilized for evaluation. The RFMiD 2.0 has 860 high-resolution color fundus images (1535x2048, 512x512, and 1000x1504) and is classified into 49 diseases by three ophthalmologists. Like RFMiD, this dataset has a pre-defined 60:40 train-test split [61,62]. To evaluate the adaptability and generalization of LiteFeatNet, a total of 372 images from the DR, MH, and NL classes were selected from RFMiD 2.0. Images from the train, validation, and test splits were combined to form a single larger test set to overcome biases associated with the number, selection, and distribution of images. Fig 10 presents the class label distribution across dataset splits for the RFMiD 2.0.

### 5.2. Dataset pre-processing

For the generalizability study the same pre-processing pipeline to the RFMiD 2.0 to make images compatible with the models discussed in the previous section. Before testing, the images from the RFMiD 2.0 dataset were converted to RGB, resized to 224x224, normalized, and converted to a format compatible with the models. The data pre-processing pipeline is depicted in Fig 11. Similarly, image labels were also encoded and converted into a model-compatible format.

### 5.3. Evaluation methodology

To evaluate various models, including LiteFeatNet, 372 images were fed to each trained model separately, which in turn generated softmax predictions for each image. Fig 12 shows the evaluation methodology.

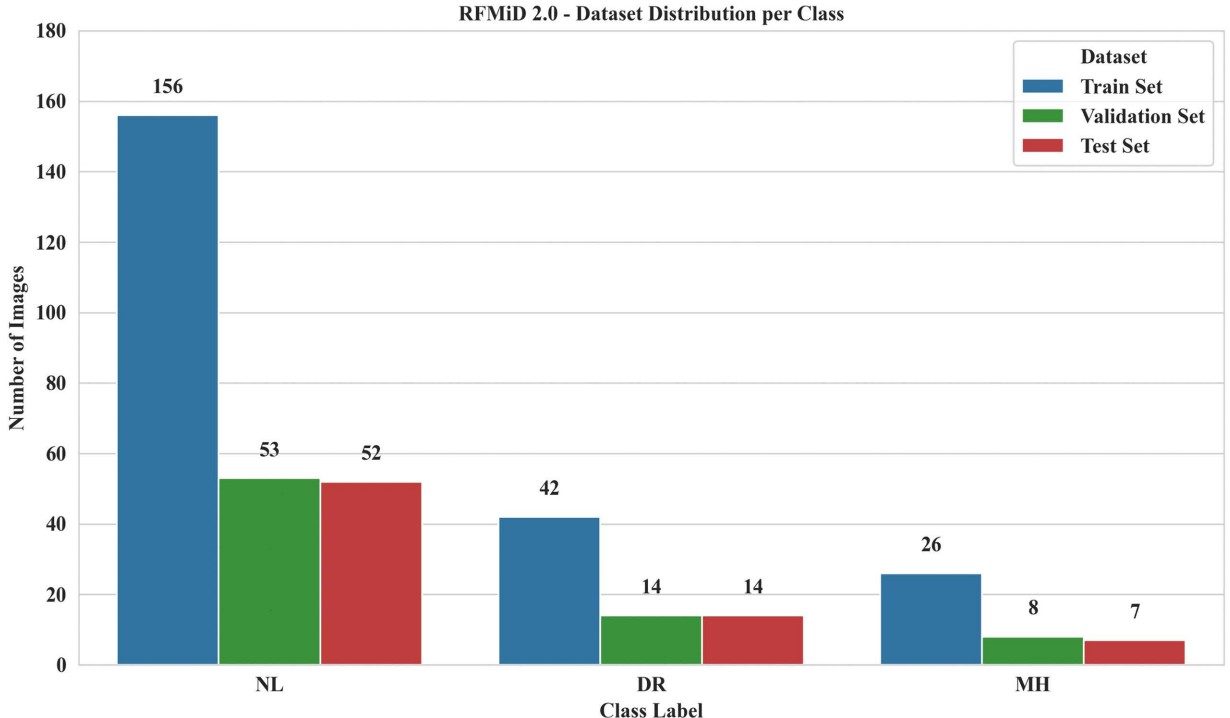

**Fig 10. Class label distribution for RFMiD 2.0.**

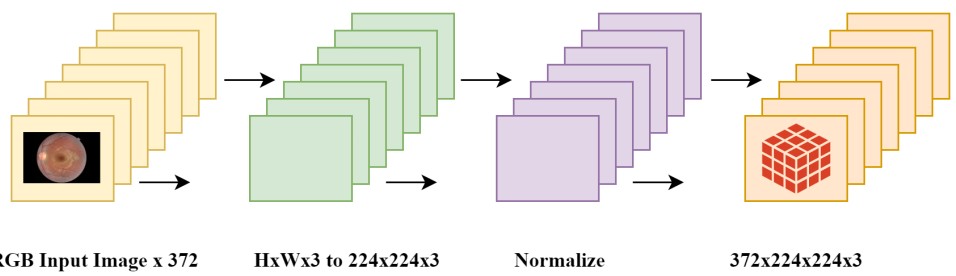

RGB Input Image x 372     HxWx3 to 224x224x3     Normalize     372x224x224x3

**Fig 11. Pre-processing pipeline for RFMiD 2.0.**

Furthermore, the entire evaluation pipeline for this study was also implemented on the Kaggle platform using the same hardware configuration as that of the training phase. To get an accurate measure of the inference time, all the input images were pre-processed and loaded into a Numpy array/Tensor completely before initiating the inference process. Similarly, all the corresponding labels were also encoded and loaded into a model-compatible format.

The test images from the RFMiD 2.0 were provided to these models independently. Each model uses its own learned weights from RFMiD, and generates output in the form of probabilities, using the softmax activation function. For each input image, the softmax prediction with highest probability was decoded into the corresponding class labels. The predicted class labels and actual class labels were used to compute evaluation metrics.

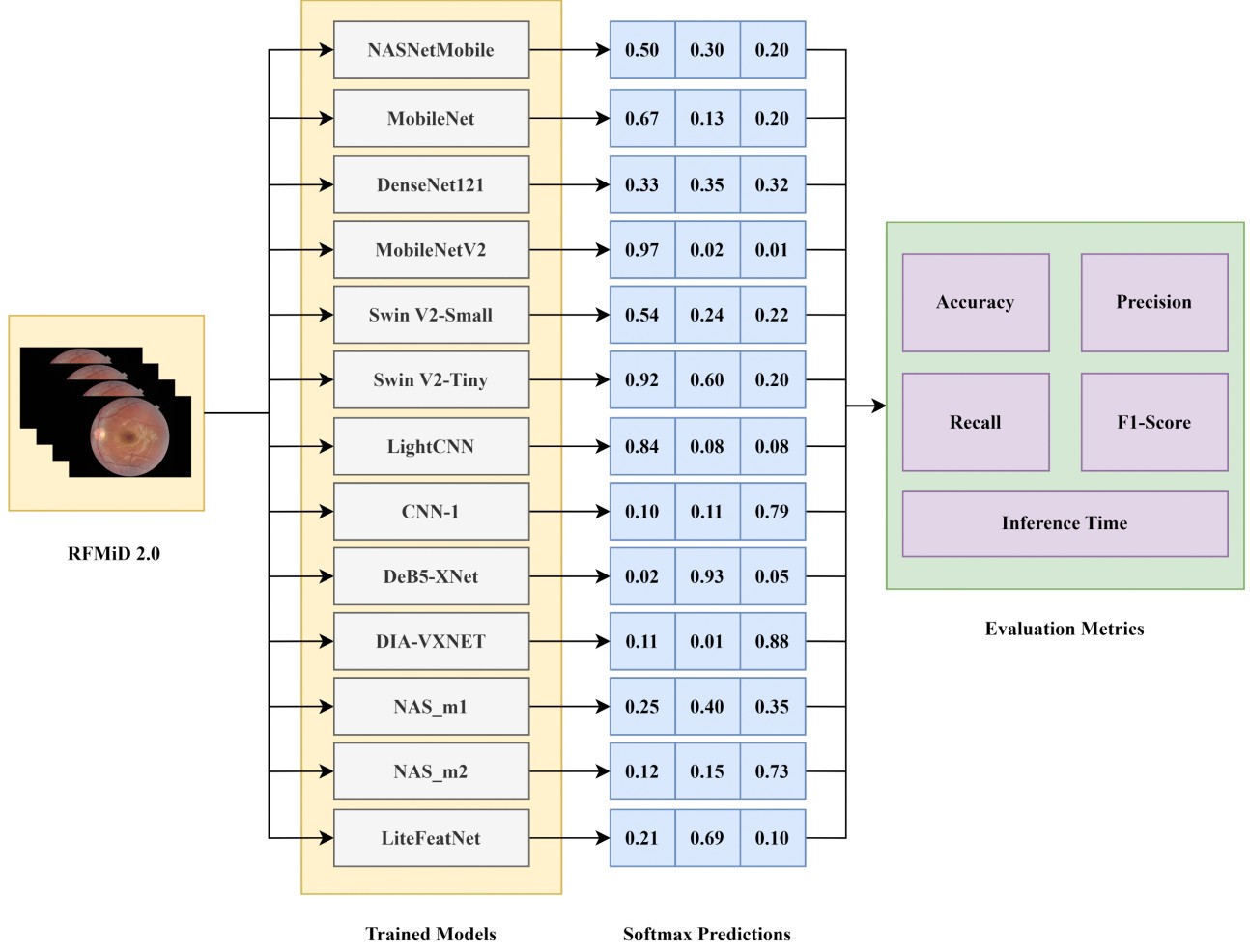

**Fig 12. Evaluation mechanism for RFMiD 2.0.**

## 5.4. Confusion matrix

Fig 13 presents the confusion matrix of LiteFeatNet's predictions for the RFMiD 2.0 corresponding to the DR, MH, and NL categories.

According to Fig 13, our LiteFeatNet model correctly classified 253 out of 261 NL, 34 out of 41 MH, and 61 out of 70 DR class instances, thereby achieving an accuracy of 93.55% on this dataset. The consistency of misclassification (DR: 9, MH: 7, and NL: 8) suggests that the proposed LiteFeatNet effectively learned the pathological patterns of all the class labels during training on RFMiD, so it exhibited a balanced generalization for RFMiD 2.0.

## 5.5. Evaluation metrics

In this section, the evaluation metrics of the benchmark models selected for this study are presented to provide detailed insights for assessing the generalizability of these models on an external dataset. Table 7 presents the evaluation metrics for various models using RFMiD 2.0.

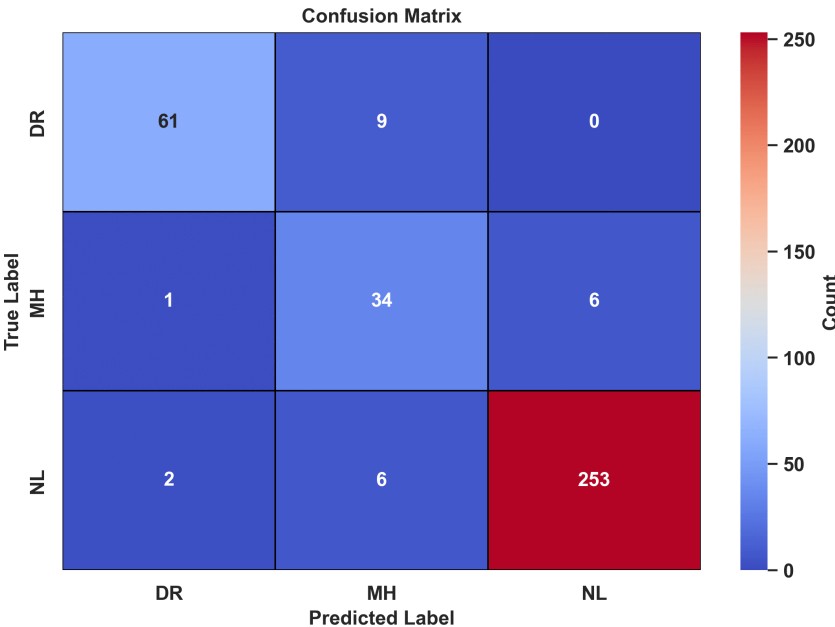

**Fig 13. Confusion matrix for LiteFeatNet using RFMiD 2.0.**

As presented in Table 7, the LiteFeatNet model achieved excellent overall performance metrics on the RFMiD 2.0. It achieved 93.55% accuracy, 94.12% precision, 93.55% recall, and 93.73% F1-score, clearly demonstrating the consistency and reliability of the evaluation metrics. Interestingly, our proposed integrated framework also reported a much faster inference time of 1.33 seconds, which is significant given the number of images utilized for inference. Among the pre-trained CNN models, DenseNet121 reported the highest accuracy (94.09%), precision (94.58%), recall (94.09%), and F1-score (94.24%). Although DenseNet121 outperformed the proposed model, its inference time (1.60 seconds) is significantly higher than that of LiteFeatNet (1.33 seconds). Thus, despite a minor performance reduction (accuracy: 0.57%, precision: 0.49%, recall: 0.57% and F1-score: 0.54%) compared with DenseNet121, our model still demonstrates deployment level efficiency (memory and time) on this external dataset.

As presented in Table 7, lightweight architectures such as MobileNet achieved competitive performance with a test time of 1.04 seconds, yet LiteFeatNet architecture outperformed it by achieving higher values of performance metrics.

The proposed LiteFeatNet also showed higher accuracy (93.55% vs. 91.67%), precision (94.12% vs. 91.13%), recall (93.55% vs. 91.67%), and F1-score (93.73% vs. 91.22%) than Swin-V2-Small. However, the Swin V2-Tiny exhibited excellent performance (accuracy: 94.89%, precision: 94.68%, recall: 94.89%, and F1-score: 94.74%), outperforming all studied models, including the proposed model. However, both Swin V2-Small and Swin V2-Tiny have much longer inference times (3.32 and 2.02 seconds), which are significantly higher than the LiteFeatNet architecture (1.33 seconds), thereby validating its superior deployment-level efficiency and predictive power.

Similarly, the accuracy (93.82%) and F1-score (93.90%) of LightCNN [16] were also marginally higher than those of the proposed model (accuracy: 93.55% and F1-score: 93.73%), as shown in Table 7. Interestingly, LiteFeatNet achieved better precision (94.12% vs. 94.04%) while requiring less inference time as compared to LightCNN [16] (1.36 seconds), suggesting a well-balanced predictive power, efficient parameter-utilization, and faster operational time. Again, CNN-1 [17] had the lowest inference time (0.93 seconds), but it demonstrated limited generalization in this domain, reporting below 90% accuracy and F1-score.

**Table 7. Evaluation metrics for RFMiD 2.0.**

| Model | Accuracy | Precision | Recall | F1-Score | Inference Time |
|---|---|---|---|---|---|
| CNN-1 [17] | 87.10 | 91.45 | 87.10 | 88.38 | 0.93 |
| DenseNet121 | 94.09 | 94.58 | 94.09 | 94.24 | 1.60 |
| DeB5-XNet [24] | 93.28 | 93.89 | 93.28 | 93.47 | 2.90 |
| DIA-VXNET [29] | 92.74 | 94.31 | 92.74 | 93.12 | 4.04 |
| LightCNN [16] | 93.82 | 94.04 | 93.82 | 93.90 | 1.36 |
| MobileNet | 92.47 | 92.77 | 92.47 | 92.47 | 1.04 |
| MobileNetV2 | 91.93 | 92.07 | 91.93 | 91.95 | 1.07 |
| NAS_m1 [30] | 92.47 | 92.92 | 92.47 | 92.57 | 1.33 |
| NAS_m2 [30] | 90.32 | 90.88 | 90.32 | 90.48 | 1.33 |
| NASNetMobile | 93.28 | 93.48 | 93.28 | 93.35 | 1.36 |
| Proposed (LiteFeatNet) | **93.55** | **94.12** | **93.55** | **93.73** | **1.33** |
| Swin V2-Small | 91.67 | 91.13 | 91.67 | 91.22 | 3.32 |
| Swin V2-Tiny | 94.89 | 94.68 | 94.89 | 94.74 | 2.02 |

Accuracy, Precision, Recall, and F1-Score (**%**), and Inference Time (seconds)

The performance evaluation metrics of ensemble architectures DeB5-XNet [24] (accuracy: 93.28% and F1-score: 93.47%) and DIA-VXNET [29] (accuracy: 92.74% and F1-score: 93.12%) was extremely close (but lower) to that of Lite-FeatNet. Interestingly, LiteFeatNet's precision (94.12%) lags that of DIA-VXNET [29] (94.31%). However, both ensemble architectures take much longer (2.90 and 4.04 seconds) to predict class labels, validating LiteFeatNet's efficiency, effectiveness, and robustness, specifically on resource-constrained environments.

Interestingly, NAS_m1 [30], NAS_m2 [30], and the proposed LiteFeatNet model use the same backbone (16.29MB non-trainable parameters from NASNetMobile) and achieve the same inference time (1.33 seconds) as that of LiteFeatNet. Still, LiteFeatNet achieved better performance compared to the other two models. Despite having 13.28M trainable parameters, the performance of NAS_m2 [30] was lower than that of the proposed model, which has 0.94M trainable parameters, suggesting that greater model complexity does not necessarily lead to better generalization. It further emphasizes the robustness of the LiteFeatNet architecture, achieving superior performance even with moderate trainable parameters.

Hence, the LiteFeatNet demonstrated robust generalization on the RFMiD 2.0, offering a superior trade-off between performance and efficiency thanks to its unique parameter optimization and compact size powered by integrating DIFE, PFMR, and the LCC modules. Its performance on RFMiD 2.0 was excellent compared with numerous state-of-the-art architectures, considering its parameter size and computational cost (inference time: 0.004 seconds/image), making it a strong prospect for practical deployment in the healthcare sector requiring efficient usage of the available resources, portable devices, and low-resourced CAD systems for effectively managing the increasing patient count.

## 6. Scalability evaluation

To further evaluate the scalability of the proposed integrated framework, we extended our original 3-class classification problem. We conducted additional experiments using increasingly complex multi-class classification scenarios with 4 and 5 class labels, respectively. To perform this extension, we added two new classes to the original 3-class dataset to introduce additional inter-class complexity, which is critical for assessing the robustness of deep learning models. For each additional disease class, we have selected a common and underrepresented retinal pathology from the RFMiD. These retinal disease classes included Macular Scar (MS) and Myopia (MYA) having 27 and 167 images in RFMiD, respectively. To

overcome the class imbalance, given our 3-class problem, we supplemented these additional categories with images from a third publicly available external dataset **(accessed on 20 February 2026)**, collected from Anawara Hamida Eye Hospital and B. N. S. B. Zahurul Haque Eye Hospital. This dataset contains 5335 images corresponding to 9 retinal pathologies [63–65]. Fig 14 shows the distribution of dataset samples across various retinal pathologies.

To upsample the underrepresented classes MS and MYA in the RFMiD, samples for these categories were extracted from this third dataset. To merge these samples with RFMiD, we split the extracted data samples into a train-validate-test ratio of 60:20:20 independently using stratified random sampling. In a recent study, [66] utilized image samples of two classes from a label-rich dataset.

The extracted samples were merged with the dataset used in our original 3-class problem. Special care was taken to prevent data leakage problem during merging, as both these datasets were collected from different geographical locations. Table 8 presents the contribution of images from both datasets for this scalability study.

We conducted numerous experiments and compared our proposed architecture with 9 state-of-the-art baseline models: Low-Cost CNN [7], CNN-2 [15], LightCNN [16], CNN-1 [17], CNN-3 [18], DeB5-XNet [24], DIA-VXNET [29], NAS_m1 [30], and NAS_m2 [30] to evaluate the scalability. It is worth mentioning that we used a consistent dataset pre-processing, augmentation pipeline, and training/evaluation setup, as in our original three class experiments.

To evaluate the performance of numerous architectures, we have considered accuracy, precision, recall, F1-score, and inference time as evaluation metrics. The overall performance metrics for the 4-class and 5-class problems are shown in Table 9 and Table 10, respectively.

As presented in Table 9, the proposed model demonstrated an accuracy (88.86%), precision (89.06%), recall (88.86%), and F1-score (88.85%), thereby making its place among the top two architectures. Among various compared architectures, DeB5-XNet [24] demonstrated the highest performance (accuracy: 89.74%, precision: 90.17%, recall: 89.74%, and F1-score: 89.72%), but at a significantly higher inference time of 3.73 seconds compared with the proposed framework. Similarly, the architectures such as LightCNN [16] and DIA-VXNET [29] also achieved competitive accuracy scores of

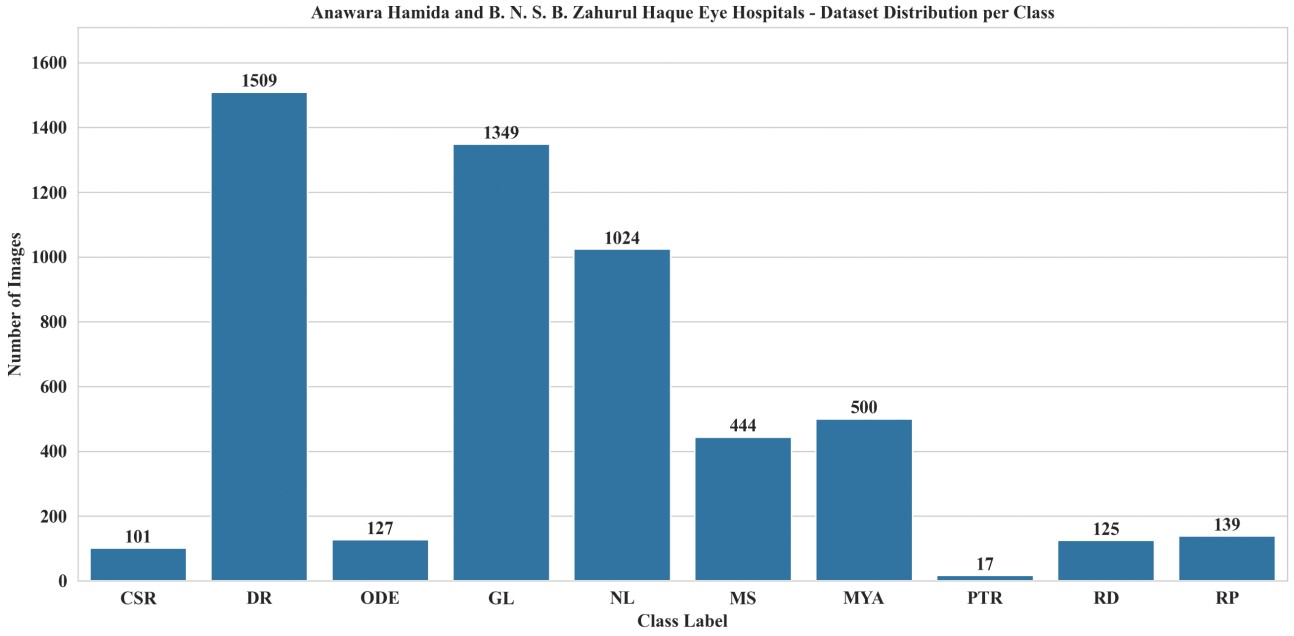

**Fig 14. External Dataset for Scalability Assessment.**

**Table 8. Class labels and the number of images supplemented for scalability assessment.**

| Multi-class problem | RFMiD | | Anawara Hamida and B. N. S. B. Zahurul Haque Eye Hospitals [63–65] | | Total Number of supplemented images |
|---|---|---|---|---|---|
| | MS | MYA | MS | MYA | |
| 4-class | 27 | – | 444 | – | 471 |
| 5-class | 27 | 167 | 444 | 500 | 1138 |

**Table 9. Performance metrics for 4 retinal pathologies.**

| Model | Accuracy | Precision | Recall | F1-Score | Inference Time |
|---|---|---|---|---|---|
| CNN-1 [17] | 78.82 | 79.66 | 78.82 | 78.67 | 1.03 |
| CNN-2 [15] | 74.02 | 75.63 | 74.02 | 73.85 | 1.05 |
| CNN-3 [18] | 82.10 | 82.60 | 82.10 | 81.99 | 0.93 |
| DeB5-XNet [24] | 89.74 | 90.17 | 89.74 | 89.72 | 3.73 |
| DIA-VXNET [29] | 87.34 | 88.03 | 87.34 | 87.26 | 5.71 |
| LightCNN [16] | 87.77 | 87.95 | 87.77 | 87.73 | 1.54 |
| Low-Cost CNN [7] | 79.04 | 79.91 | 79.04 | 79.06 | 4.40 |
| NAS_m1 [30] | 84.28 | 84.46 | 84.28 | 84.27 | 1.58 |
| NAS_m2 [30] | 83.84 | 83.40 | 83.84 | 83.79 | 1.61 |
| Proposed (LiteFeatNet) | **88.86** | **89.06** | **88.86** | **88.85** | **1.55** |

Accuracy, Precision, Recall, and F1-Score (%), and Inference Time (seconds).

**Table 10. Performance metrics for 5 retinal pathologies.**

| Model | Accuracy | Precision | Recall | F1-Score | Inference Time |
|---|---|---|---|---|---|
| CNN-1 [17] | 70.51 | 72.00 | 70.51 | 70.32 | 1.31 |
| CNN-2 [15] | 65.59 | 68.07 | 65.59 | 65.14 | 1.32 |
| CNN-3 [18] | 73.39 | 76.78 | 73.39 | 73.29 | 1.21 |
| DeB5-XNet [24] | 87.46 | 87.97 | 87.46 | 87.33 | 4.44 |
| DIA-VXNET [29] | 87.63 | 87.85 | 87.63 | 87.56 | 8.28 |
| LightCNN [16] | 85.08 | 85.87 | 85.08 | 85.18 | 1.97 |
| Low-Cost CNN [7] | 74.24 | 74.94 | 74.24 | 74.22 | 5.71 |
| NAS_m1 [30] | 80.85 | 81.41 | 80.85 | 80.97 | 1.93 |
| NAS_m2 [30] | 84.74 | 85.10 | 84.74 | 84.79 | 1.93 |
| Proposed (LiteFeatNet) | **85.42** | **86.52** | **85.42** | **85.53** | **1.95** |

Accuracy, Precision, Recall, and F1-Score (%), and Inference Time (seconds).

87.77% and 87.34%, respectively with DIA-VXNET [29] having the highest inference time of 5.71 seconds among all baseline architectures. In contrast, the LiteFeatNet exhibited the second-best performance metrics, with a considerably lower testing time of 1.55 seconds compared with DeB5-XNet [24], LightCNN [16], and DIA-VXNET [29]. Our proposed model balances the classification performance and computational efficiency compared with NASNetMobile-based light-weight architectures NAS_m1 [30] and NAS_m2 [30]. On the other hand, the CNN architectures such as CNN-1 [17], CNN-2 [15], CNN-3 [18], and Low-Cost CNN [7], despite having much lower inference time, achieved considerably lower performance with accuracy scores ranging from 74% to 82%. This clearly indicates that these architectures struggled to

learn pathological features for this classification problem with four distinct categories. In contrast, the LiteFeatNet architecture captured disease-relevant features effectively and efficiently through the synergistic combination of DIFE, PFMR and LCC modules.

Table 10 summarizes the performance of various architectures for another larger multi-class scenario having 5 categories. According to Table 10, architectures such as CNN-1 [17], CNN-2 [15], and CNN-3 [18] experienced substantial performance drops from 4-class scenario, with accuracy scores of 70.51%, 65.59%, and 73.39%, respectively. This performance once again highlights the limitation of these architectures in capturing complex pathological features. On the other hand, the architectures NAS_m2 [30], NAS_m1 [30], and Low-Cost CNN [7] improved classification performance, achieving accuracies of 84.74%, 80.85%, and 74.24%, respectively. Despite the lower inference time (1.93 seconds), both NAS_m2 [30] and NAS_m1 [30] underperformed the top 3 best-performing architectures: DIA-VXNET [29], DeB5-XNet [24], and the proposed LiteFeatNet architecture.

The architectures DIA-VXNET [29] and DeB5-XNet [24], being top-ranked, demonstrated strong discriminative capabilities with performance (accuracy: 87.63% vs. 87.46%, precision: 87.85% vs. 87.97%, recall: 87.63% vs. 87.46%, and F1-score: 87.56% vs. 87.33%). However, both these architectures were computationally costly with higher inference times (8.28 vs. 4.44 seconds). This increased testing time may limit their real-time clinical implementation, particularly in scenarios with higher patient count, which are common worldwide. In comparison, LiteFeatNet successfully maintained competitive predictive power, achieving an accuracy of 85.42%, precision of 86.52%, recall of 85.42%, and F1-score of 85.53%, while demonstrating a low computational overhead with 1.95 seconds as inference time. Moreover, the performance degradation of the LiteFeatNet model from a 4-class to a 5-class scenario was minimal, thereby validating the robustness and scalability of the proposed LiteFeatNet, even with increasing class label complexity.

Overall, LiteFeatNet demonstrated robust predictive performance and outperformed numerous state-of-the-art baseline models, thereby indicating its superior feature extraction and learning. Moreover, its minimal performance reduction, when scaling from a 4-class to a 5-class scenario, highlights LiteFeatNet's generalizability to more diverse label spaces. The results also validate that the integration of the architectural components (DIFE, PFMR, and LCC) has enabled our integrated framework to be robust, scalable, adaptable, and computationally efficient, beyond the original 3-class experimental scenario, thereby positioning it as a viable automated diagnostic tool suitable for larger multi-class problems in real-world clinical deployment.

## 7. Ablation study

To evaluate the individual and combined impact of DIFE and the PFMR module, a comprehensive ablation study is also conducted using the RFMiD. To obtain detailed insights into the integration of DIFE and PFMR, we conducted ablation experiments for our proposed framework and two backbone architectures.

### 7.1. Ablation experiments for the proposed framework

To understand the impact of integrating DIFE and PFMR in our proposed architecture, we conducted ablation experiments using four different model variants (model title suffixes: A, B, C, and D) of our proposed architecture. The evaluation metrics (accuracy, precision, recall, and F1-score), training, and inference time for four different versions of the proposed architecture: (1) the full LiteFeatNet that incorporates both DIFE and PFMR (alias: LiteFeatNetA for the ablation study only), (2) a baseline model without both components (referred as LiteFeatNetB), (3) a model using only DIFE (mentioned as LiteFeatNetC), and (4) a model using only PFMR (called as LiteFeatNetD) were compared. The comparison of training and inference times assisted in assessing the impact of the DIFE and PFMR on the time complexity of the LiteFeatNet model. Table 11 summarizes the evaluation metrics, training, and testing time for the various ablation experiments.

According to Table 11, the first ablation experiment with the complete model LiteFeatNetA achieved the highest accuracy of 90.33%, precision of 90.69%, recall of 90.33%, and F1-score of 90.27%. The evaluation metrics highlight the

**Table 11. Ablation study experiments for the proposed LiteFeatNet.**

| Ablation Model | DIFE | PFMR | Accuracy | Precision | Recall | F1-Score | Training Time | Inference Time |
|---|---|---|---|---|---|---|---|---|
| LiteFeatNetA | ✓ | ✓ | 90.33 | 90.69 | 90.33 | 90.27 | 409.35 | 1.32 |
| LiteFeatNetB | ✗ | ✗ | 87.57 | 87.49 | 87.57 | 87.49 | 428.03 | 1.27 |
| LiteFeatNetC | ✓ | ✗ | 88.12 | 88.17 | 88.12 | 88.04 | 416.53 | 1.30 |
| LiteFeatNetD | ✗ | ✓ | 89.50 | 89.50 | 89.50 | 89.45 | 421.58 | 1.32 |

Accuracy, Precision, Recall, and F1-Score (%), Training Time, and Inference Time (seconds)

synergic effect of combining DIFE with PFMR. Integrating both techniques enabled the model to capture both low-level and high-level discriminative features from fundus images across different class labels.

In contrast to LiteFeatNetA, the second ablation experiment with LiteFeatNetB model demonstrated a noticeable drop in the overall performance, yielding a lower accuracy (87.57%), precision (87.49%), recall (87.57%), and F1-score (87.49%). This highlights the importance of integrating both modules in enhancing the model's learning capability and predictive power.

The LiteFeatNetC model improved performance slightly over LiteFeatNetB model, achieving 88.12% accuracy and 88.04% F1-score, highlighting the effectiveness of deep feature extraction before NASNetMobile's ReLU activation. It also implies that intermediate features do contribute to better feature representation; however, their impact is subtle when applied without PFMR. Interestingly, the LiteFeatNetD outperformed the LiteFeatNetC, achieving accuracy (89.50%) and F1-score (89.45%) quite close to those of the proposed LiteFeatNet, effectively highlighting the significance of integrating the PFMR module for performance improvement. It also suggests that the PFMR does not solely reduce the feature channels but also confirms its ability to filter refined feature maps by suppressing redundant and irrelevant features, thereby reducing the time required for the training process.

Nevertheless, the LiteFeatNetA derives its substantial benefits and predictive strength from the synergistic combination of DIFE and PFMR, confirming their complementary strengths and roles in designing a memory- and time-efficient, reliable deep neural network architecture for classifying multiple retinal pathologies effectively. The comparison of evaluation metrics used for various experiments in the ablation study for LiteFeatNet is summarized in Fig 15 for better visualization.

According to Table 11, LiteFeatNetB took the longest training time (428.03 seconds) among all variants, possibly due to inefficient weight updates in the absence of a feature refinement strategy, emphasizing the need for more epochs for improving performance. This configuration takes 1.27 seconds to generate an inference for 362 test images, which is lowest among other ablation experiments with the proposed framework.

The LiteFeatNetD model reduced the training time of the LiteFeatNetB model from 428.03 seconds to 421.58 seconds, while the testing time increased to 1.32 seconds. Despite having slightly higher computational overhead, the PFMR provided the model with a competitive advantage by generating relevant feature maps. On the other hand, the LiteFeatNetC model significantly reduced the training time of the baseline configuration to 416.53 seconds, and it also exhibited a testing time of 1.30 seconds. The LiteFeatNetC model suggests that intermediate features assisted the model by providing meaningful features extracted early in the backbone, thereby accelerating inference and training.

The LiteFeatNetA model slightly increased the testing time to 1.32 seconds, which is fully justified by the integration of DIFE, PFMR and LCC modules. However, this slight rise is fully compensated by superior classification performance, resulting in computationally efficient operations, particularly considering lower inference and training times. Surprisingly, the training time (409.35 seconds) was significantly lower than that of all other versions, strongly suggesting that despite adding the PFMR module and associated slight computational overhead, the model was still able to converge faster.

Hence, this ablation study strongly validates that both DIFE and PFMR contribute implicitly to achieving the optimal classification performance while maintaining computational efficiency. On the whole, the integration of DIFE and PFMR

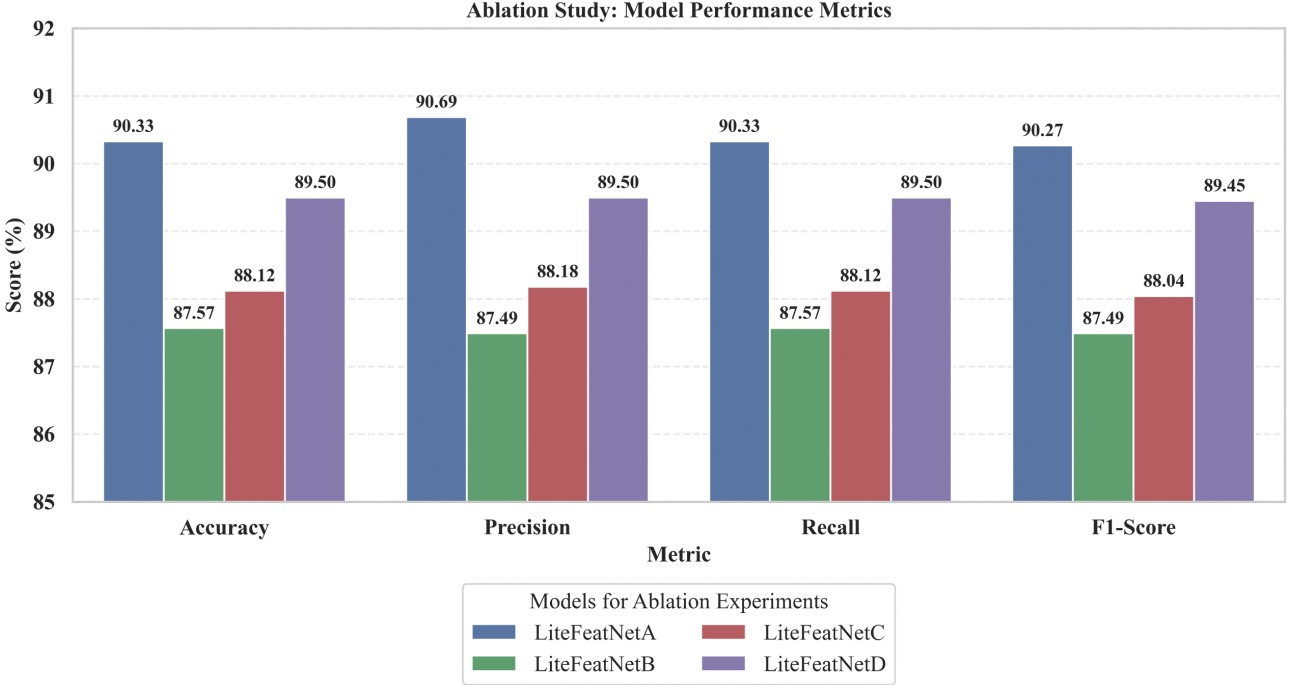

**Fig 15. Performance evaluation of ablation experiments.**

modules made LiteFeatNet outperformed all state-of-the-art transfer learning methods and well-established architectures across all performance metrics, making it efficient and robust enough to avoid misdiagnosis in healthy patients.

### 7.2. Ablation experiments for other backbone architectures

We also evaluated the impact of DIFE and PFMR with two top-performing backbone feature extraction architectures: DenseNet121 and MobileNetV2. To conduct these ablation experiments, we evaluated eight model variants, four for each DenseNet121 and MobileNetV2 backbone feature extractor. The suffixes (A, B, C, and D) denote the consistency in ablation model configurations. The architectural specifications used in these ablation experiments and corresponding evaluation metrics for this second set of experiments are shown in Table 12.

**Table 12. Ablation experiments for other backbone architectures.**

| Ablation Model | DIFE | PFMR | Accuracy | Precision | Recall | F1-Score | Training Time | Inference Time |
|---|---|---|---|---|---|---|---|---|
| DenseNet121A | ✓ | ✓ | 89.78 | 90.21 | 89.78 | 89.71 | 401.93 | 1.57 |
| DenseNet121B | ✗ | ✗ | 88.40 | 89.25 | 88.40 | 88.33 | 420.59 | 1.46 |
| DenseNet121C | ✓ | ✗ | 89.23 | 90.00 | 89.23 | 89.15 | 401.31 | 1.43 |
| DenseNet121D | ✗ | ✓ | 89.50 | 89.87 | 89.50 | 89.44 | 413.68 | 1.53 |
| MobileNetV2A | ✓ | ✓ | 86.19 | 86.61 | 86.19 | 86.02 | 350.62 | 1.00 |
| MobileNetV2B | ✗ | ✗ | 83.98 | 84.58 | 83.98 | 83.72 | 343.02 | 0.98 |
| MobileNetV2C | ✓ | ✗ | 85.36 | 86.16 | 85.36 | 85.11 | 345.75 | 0.98 |
| MobileNetV2D | ✗ | ✓ | 86.19 | 86.57 | 86.19 | 86.01 | 361.12 | 1.01 |

Accuracy, Precision, Recall, and F1-Score (%), Training Time, and Inference Time (seconds)

To provide a quantitative measure of the effectiveness of the DIFE, PFMR, and LCC modules, we have evaluated the improvement (%) in the performance of various ablation models by using their corresponding transfer learning architectures as baseline references. The percentage improvement/reduction in the performance of NASNetMobile, DenseNet121, and MobileNetV2, are presented in Fig 16, Fig 17, and Fig 18, respectively.

As shown in Fig 16, the LiteFeatNetB improved the accuracy, precision, recall, and F1-score by 0.96%, 0.91%, 0.96%, and 0.93%, respectively, compared with the reference fine-tuned NASNetMobile. Conversely, the same architectural configuration with DenseNet121 and MobileNetV2 backbones reported considerable performance reduction (accuracy: 0.93%, precision: 0.49%, recall: 0.93%, and F1-score: 0.88%) and (accuracy: 1.29%, precision: 1.18%, recall: 1.29%, and F1-score: 1.37%), as shown in Fig 17 and Fig 18, respectively, in comparison to the baseline transfer learning settings. These results effectively highlight the role of the LCC module in the performance improvement of the proposed framework. It is also revealed that integrating the LCC module solely with DenseNet121 and MobileNetV2 backbones results in performance drop, thereby limiting its effectiveness with these architectures, particularly when integrated alone.

As shown in Fig 16, the LiteFeatNetC model improved performance (accuracy: 1.59%, precision: 1.70%, recall: 1.59%, and F1-score: 1.57%) in comparison with the reference baseline. Whereas, according to Fig 17, DenseNet121C marginally increased baseline's precision and F1-score of the by 0.35% and 0.04%, respectively. The MobileNetV2C, as shown in Fig 18, also reported a fractional performance improvement of 0.33% in accuracy, 0.67% in precision, 0.33% in recall, and 0.27% in F1-score, as compared with MobileNetV2. Thus, integration of the DIFE proved more beneficial for LiteFeatNet than the DenseNet121 and MobileNetV2 backbones, which again validates our design decision behind the proposed framework.

The model LiteFeatNetD improved the performance metrics of LiteFeatNetC by 2-fold (approximately), as shown in Fig 16. It achieved improvements of 3.18% in accuracy, 3.23% in precision, 3.18% in recall, and 3.20% in F1-score, compared with the NASNetMobile. As shown in Fig 17, DenseNet121D slightly improved all the performance metrics (accuracy: 0.30%, recall: 0.30%; precision: 0.20%; F1-score: 0.37%) compared to the baseline DenseNet121. Whereas, the configuration MobileNetV2D exhibited substantial improvement across all performance evaluation metrics (accuracy and recall: 1.30%; F1-score: = 1.33%; precision: 1.14%), as given in Fig 18, clearly validating the effectiveness of the PFMR module, particularly for lightweight architectures such as NASNetMobile and MobileNetV2.

Interestingly, all the DIFE-PFMR-integrated model variants, LiteFeatNetA, DenseNet121A, and MobileNetV2A, dramatically improved the performance of the corresponding baseline architectures. According to Fig 16, the performance of the NASNetMobile baseline was improved by a comprehensive margin (accuracy: 4.14%, precision: 4.60%, recall: 4.14%,

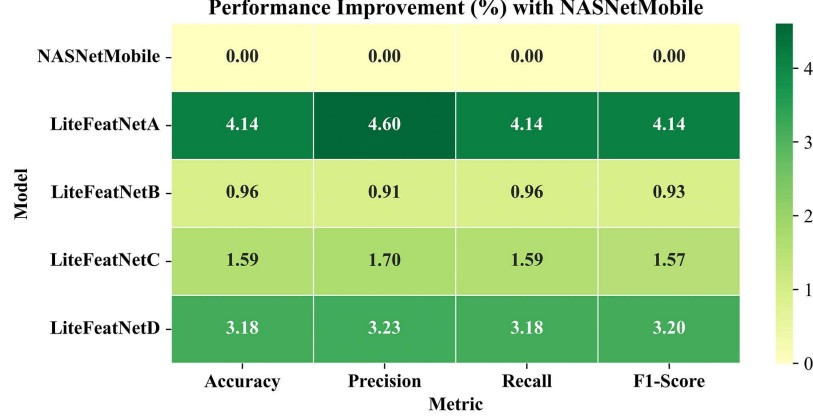

**Fig 16. Performance improvement with NASNetMobile backbone.**

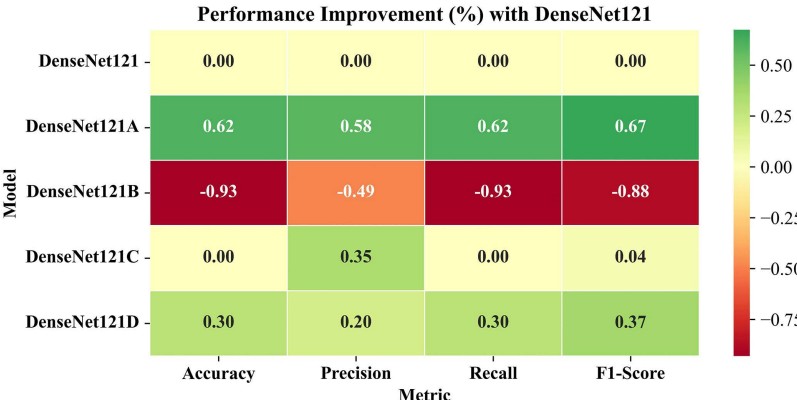

**Fig 17. Performance improvement with DenseNet121 backbone.**

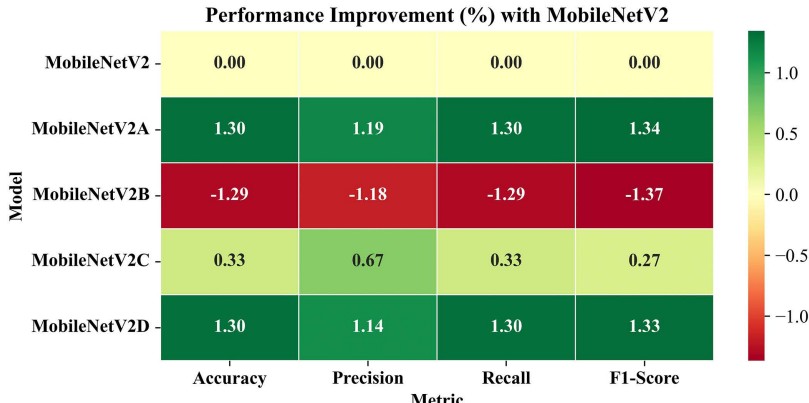

**Fig 18. Performance improvement with MobileNetV2 backbone.**

and F1-score: 4.14%). Similarly, DenseNet121A also showed moderate improvements in accuracy (0.62%), precision (0.58%), recall (0.62%), and F1-score (0.67%), as shown in Fig 17. MobileNetV2A, though reported only a marginal increase in precision and F1-score compared with MobileNetV2D, demonstrated a substantial improvement of 1.30% in accuracy, 1.19% in precision, 1.30% in recall, and 1.34% in F1-score (as presented in Fig 18) compared with the baseline version. The performance metrics achieved confirm that the effectiveness of integrating DIFE and PFMR module across different backbones was superior when compared with other configurations.

Fig 19 compares the inference times of LiteFeatNetA, DenseNet121A, and MobileNetV2A with their counterparts. According to Fig 19, the ablation models MobileNetV2A, DenseNet121A, and LiteFeatNetA reported lower inference times (1.00, 1.57, and 1.32 seconds, respectively) than those of MobileNetV2, DenseNet121, and NASNetMobile (1.05, 1.59, and 1.34 seconds, respectively). This, once again confirms that the DIFE and PFMR synergy empowers transfer learning models to be computationally efficient, thereby enabling deployment-level efficiency in resource-constrained environments.

All results with the configuration utilizing both DIFE and PFMR confirm the synergistic effect of DIFE and PFMR for performance improvement across all the baselines, in particular. Moreover, their integration with NASNetMobile backbone

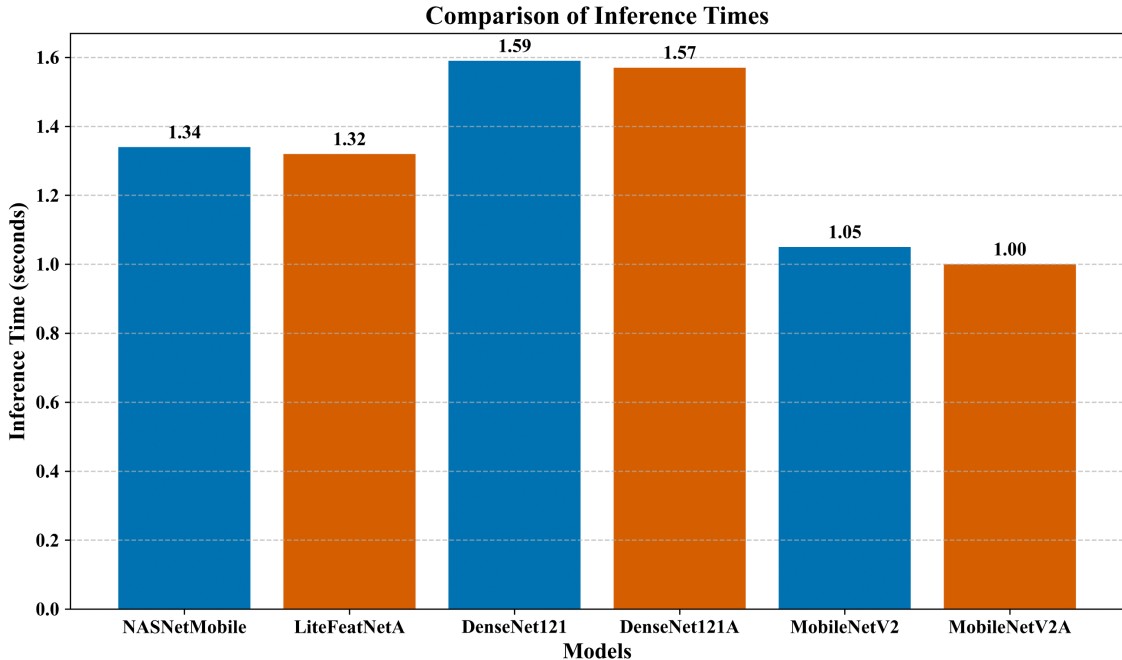

**Fig 19. Comparison of inference times.**

demonstrated the effectiveness, efficiency, and motivation behind the specific architectural choices made for designing a lightweight, performance-centric, and computationally efficient LiteFeatNet architecture.

## 8. Conclusions

Designing lightweight models for low-resource environments is one of the key challenges for countries with limited health-care resources and personnel. Therefore, a parameter-efficient deep learning model was proposed that leverages transfer learning, utilizing NASNetMobile as the backbone feature extractor. The LiteFeatNet model achieved superior performance compared to numerous standard deep learning architectures due to the incorporation of DIFE, a spatial-loss-aware feature refinement module (PFMR), and a compact 6-layer LCC module.

The proposed deep learning-based integrated framework demonstrated excellent classification performance across all DR, MH, and NL class labels. Moreover, it also exhibited robustness and generalization on the RFMiD 2.0. Meanwhile, the LiteFeatNet model significantly reduced the number of parameters, associated model size, and computational efficiency, particularly the inference time. It also outperformed deeper and heavier models in every respect, validating the effectiveness and computational efficient of the proposed LiteFeatNet model. The ablation study also validated the synergistic effect of DIFE and PFMR, identifying it as critical for achieving optimal performance, resource, and computational efficiency from the proposed architecture.

The findings from experiments suggest that LiteFeatNet, powered by memory-optimized and time-efficient low-learning, has proven that it can maintain higher accuracy and sensitivity without sacrificing sensitivity, making it a suitable choice for deployment in real-time, resource-constrained environments, particularly in underdeveloped regions, where both diagnostic reliability and computational resource efficiency are incredibly crucial.

Although LiteFeatNet achieved the best performance metrics among numerous models used in this study, there is still room for further optimization and experimentation. Reducing the resource requirements of the model and enhancing its predictive performance remain open horizons for developing robust and computationally efficient deep learning models.

## Acknowledgments

We want to express heartfelt appreciation to everyone who offered motivation during the preparation of this manuscript. Their feedback and appreciation have been instrumental in improving the quality of this research.

## Author contributions

**Conceptualization:** Usman Rafi, Qamar Nawaz, Aisha Khatoon.

**Data curation:** Usman Rafi.

**Formal analysis:** Usman Rafi, Qamar Nawaz, Muhammad Ahsan Latif.

**Investigation:** Usman Rafi, Qamar Nawaz.

**Methodology:** Usman Rafi, Qamar Nawaz, Muhammad Ahsan Latif, Aisha Khatoon.

**Project administration:** Qamar Nawaz.

**Resources:** Qamar Nawaz, Muhammad Ahsan Latif, Aisha Khatoon.

**Software:** Usman Rafi, Qamar Nawaz.

**Supervision:** Qamar Nawaz, Muhammad Ahsan Latif.

**Validation:** Usman Rafi, Qamar Nawaz, Muhammad Ahsan Latif, Aisha Khatoon.

**Visualization:** Usman Rafi, Muhammad Ahsan Latif, Aisha Khatoon.

**Writing – original draft:** Usman Rafi.

**Writing – review & editing:** Usman Rafi, Qamar Nawaz, Muhammad Ahsan Latif, Aisha Khatoon.

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
