## [Decision Letter · Decision Letter 0]

17 Oct 2025

PONE-D-25-44797LiteFeatNet: A parameter-efficient and performance-centric deep learning model for multi-ocular disease identification using intermediate feature reduction from fundus imagesPLOS ONE

Dear Dr. RAFI,

Thank you for submitting your manuscript to PLOS ONE. After careful consideration, we feel that it has merit but does not fully meet PLOS ONE’s publication criteria as it currently stands. Therefore, we invite you to submit a revised version of the manuscript that addresses the points raised during the review process.

We look forward to receiving your revised manuscript.

Kind regards,

Burak Tasci, Ph.D.

Academic Editor

PLOS ONE

**Journal Requirements:**

1. When submitting your revision, we need you to address these additional requirements. Please ensure that your manuscript meets PLOS ONE's style requirements, including those for file naming. The PLOS ONE style templates can be found at https://journals.plos.org/plosone/s/file?id=wjVg/PLOSOne_formatting_sample_main_body.pdf and https://journals.plos.org/plosone/s/file?id=ba62/PLOSOne_formatting_sample_title_authors_affiliations.pdf 2. Please note that PLOS One has specific guidelines on code sharing for submissions in which author-generated code underpins the findings in the manuscript. In these cases, we expect all author-generated code to be made available without restrictions upon publication of the work. Please review our guidelines at https://journals.plos.org/plosone/s/materials-and-software-sharing#loc-sharing-code and ensure that your code is shared in a way that follows best practice and facilitates reproducibility and reuse. 3. Thank you for stating the following in the Acknowledgments Section of your manuscript: No funding was received for this study. We note that you have provided funding information that is not currently declared in your Funding Statement. However, funding information should not appear in the Acknowledgments section or other areas of your manuscript. We will only publish funding information present in the Funding Statement section of the online submission form. Please remove any funding-related text from the manuscript and let us know how you would like to update your Funding Statement. Currently, your Funding Statement reads as follows: The author(s) received no specific funding for this work.  Please include your amended statements within your cover letter; we will change the online submission form on your behalf. 4. Thank you for uploading your study's underlying data set. Unfortunately, the repository you have noted in your Data Availability statement does not qualify as an acceptable data repository according to PLOS's standards. At this time, please upload the minimal data set necessary to replicate your study's findings to a stable, public repository (such as figshare or Dryad) and provide us with the relevant URLs, DOIs, or accession numbers that may be used to access these data. For a list of recommended repositories and additional information on PLOS standards for data deposition, please see https://journals.plos.org/plosone/s/recommended-repositories. 5. If the reviewer comments include a recommendation to cite specific previously published works, please review and evaluate these publications to determine whether they are relevant and should be cited. There is no requirement to cite these works unless the editor has indicated otherwise.

**Additional Editor Comments:**

Thank you for submitting your manuscript to PLOS ONE. Following editorial and peer review, several revisions are required to ensure that your work meets the journal’s standards for transparency, scientific rigor, and clarity. First, in alignment with PLOS ONE’s commitment to open science and reproducibility, we kindly request that you make the code used in your study publicly available via a trusted platform such as GitHub, GitLab, or Zenodo. Please include the corresponding access link in the Data Availability Statement of your manuscript.

Additionally, we recommend a thorough revision of the manuscript to improve the quality of the English language. This includes addressing grammatical errors, improving sentence structure, and ensuring overall clarity. You may consider using a professional editing service to ensure the language meets academic publishing standards.

We also encourage you to update your literature review by incorporating recent and relevant studies from the past few years. This will help contextualize your work within current research trends and reinforce the relevance of your findings.

Lastly, please prepare a detailed response letter addressing each of the reviewers' comments. For every point raised, clearly explain how it was addressed and indicate where the corresponding changes were made in the manuscript. If there are suggestions you choose not to implement, please provide a reasoned explanation. You are not obligated to include or cite any references suggested by reviewers if they are not directly relevant to your study. We encourage you to critically evaluate each suggestion and include only those that add value to your work.

We appreciate your efforts in improving the manuscript and look forward to receiving your revised submission.

Reviewers' comments:

Reviewer's Responses to Questions

**Comments to the Author**

1. Is the manuscript technically sound, and do the data support the conclusions?

Reviewer #1: Yes

Reviewer #2: Yes

2. Has the statistical analysis been performed appropriately and rigorously? 

Reviewer #1: Yes

Reviewer #2: Yes

3. Have the authors made all data underlying the findings in their manuscript fully available?

Reviewer #1: Yes

Reviewer #2: Yes

4. Is the manuscript presented in an intelligible fashion and written in standard English?

Reviewer #1: Yes

Reviewer #2: Yes

5. Review Comments to the Author

**Reviewer #1:** The work addresses an important issue of designing and developing a lightweight, efficient, and reliable deep learning model for accurate detection and identification of multiple retinal conditions. It provides valuable insights that will be of interest to the readership of this journal. The overall structure is logical, the writing is clear, and the results are well presented. I recommend the acceptance.

**Reviewer #2:** This paper introduces a robust CNN 19 (called LiteFeatNet) that requires fewer trainable parameters, computational 20 resources, and processing time for accurate prediction. To enhance the robustness of 21 the learning process and reduce computational time, pre-trained NASNetMobile is 22 employed as the backbone and Deep Intermediate Feature Extraction (DIFE) is 23 proposed for extracting discriminative features. The extracted features are refined 24 through a novel spatially-aware Pointwise Feature Map Reduction (PFMR) module, 25 and classified using a custom classification module with a reduced number of 26 trainable parameters, computational resources, and processing time. Experiments are 27 conducted using 1824 images belonging to three distinct class labels from the Retinal 28 Fundus Multi-Disease Image Dataset (RFMiD), with a 60:40 train-test split. The 29 proposed architecture has a compact size (19.87 MegaBytes or MB) and it 30 outperformed fourteen state-of-the-art models by achieving the highest accuracy 31 (90.33%), precision (90.69%), recall (90.33%), and F1-score (90.27%), after 6.82 32 minutes (6 minutes and 49.35 seconds) of training with a standard pre-processing 33 pipeline and training configurations. Further, a generalizability study of the proposed 34 model was also conducted with the help of an external dataset, called RFMiD 2.0, and 35 the proposed architecture achieved competitive performance with quicker testing time 36 compared to other architectures. An ablation study was also conducted to validate that 37 the combination of Deep Intermediate Feature Extraction (DIFE) and Pointwise 38 Feature Map Reduction (PFMR) is the major design decision behind the success of 3 39 the LiteFeatNet architecture. The evaluation metrics suggest that the proposed 40 architecture is a lightweight, fast, and robust, making it highly suitable for 41 development and real-time operations on low-end computing devices such as 42 smartphones.

Good work keeps up

But some comments are needed?

all the comments with editor

6. PLOS authors have the option to publish the peer review history of their article (what does this mean?). If published, this will include your full peer review and any attached files.

Reviewer #1: No

Reviewer #2: No

---

## [Author Response · Author response to Decision Letter 1]

27 Oct 2025

Response to Editor Comments:

We appreciate the editor’s valuable feedback and suggestions. We have carefully revised the paper in accordance with the Editor’s comments and suggestions. All modifications have been incorporated to improve the clarity and quality of the manuscript.

Response to Reviewer 1:

We sincerely thank the Reviewer 1 for their time and effort in reviewing our manuscript.

Response to Reviewer 2:

We sincerely thank the Reviewer 2 for their time and effort in reviewing our manuscript.

---

## [Decision Letter · Decision Letter 1]

19 Feb 2026

PONE-D-25-44797R1LiteFeatNet: A parameter-efficient and performance-centric deep learning model for multi-ocular disease identification using intermediate feature reduction from fundus imagesPLOS One

Dear Dr. RAFI,

Thank you for submitting your manuscript to PLOS ONE. After careful consideration, we feel that it has merit but does not fully meet PLOS ONE’s publication criteria as it currently stands. Therefore, we invite you to submit a revised version of the manuscript that addresses the points raised during the review process.

I would like to thank you for submitting your manuscript to PLOS ONE.

The topic addressed in this study is timely and relevant, and the manuscript presents a technically sound approach with meaningful experimental validation. The methodology is generally well described, and the results appear promising. I believe the paper has merit and can be considered for publication after minor revisions.

However, several points should be clarified or slightly improved to enhance the clarity and rigor of the manuscript.

We look forward to receiving your revised manuscript.

Kind regards,

Taikyeong Ted Jeong, Ph.D.

Academic Editor

PLOS One

**Journal Requirements:**

**Additional Editor Comments:**

I would like to thank you for submitting your manuscript to PLOS ONE.

The topic addressed in this study is timely and relevant, and the manuscript presents a technically sound approach with meaningful experimental validation. The methodology is generally well described, and the results appear promising. I believe the paper has merit and can be considered for publication after minor revisions.

However, several points should be clarified or slightly improved to enhance the clarity and rigor of the manuscript.

Reviewers' comments:

Reviewer's Responses to Questions

**Comments to the Author**

1. If the authors have adequately addressed your comments raised in a previous round of review and you feel that this manuscript is now acceptable for publication, you may indicate that here to bypass the “Comments to the Author” section, enter your conflict of interest statement in the “Confidential to Editor” section, and submit your "Accept" recommendation.

Reviewer #2: All comments have been addressed

Reviewer #3: (No Response)

2. Is the manuscript technically sound, and do the data support the conclusions?

Reviewer #2: Yes

Reviewer #3: Partly

3. Has the statistical analysis been performed appropriately and rigorously? 

Reviewer #2: Yes

Reviewer #3: N/A

4. Have the authors made all data underlying the findings in their manuscript fully available?

Reviewer #2: Yes

Reviewer #3: No

5. Is the manuscript presented in an intelligible fashion and written in standard English?

Reviewer #2: Yes

Reviewer #3: Yes

6. Review Comments to the Author

**Reviewer #2:** This study introduces a robust CNN

(called LiteFeatNet) that requires fewer trainable parameters, computational resources,

and processing time for accurate prediction. To enhance the robustness of the learning

process and reduce computational time, pre-trained NASNetMobile is employed as the

backbone and Deep Intermediate Feature Extraction (DIFE) is proposed for extracting

discriminative features. The extracted features are refined through a novel spatiallyaware Pointwise Feature Map Reduction (PFMR) module, and classified using a

custom classification module with a reduced number of trainable parameters,

computational resources, and processing time. Experiments are conducted using 1824

images belonging to three distinct class labels from the Retinal Fundus Multi-Disease

Image Dataset (RFMiD), with a 60:40 train-test split. The proposed architecture has a

compact size (19.87 MegaBytes or MB) and it outperformed fourteen state-of-the-art

models by achieving the highest accuracy (90.33%), precision (90.69%), recall

(90.33%), and F1-score (90.27%), after 6.82 minutes (6 minutes and 49.35 seconds) of

training with a standard pre-processing pipeline and training configurations

good work

**Reviewer #3:** This manuscript proposes LiteFeatNet, a lightweight convolutional neural network for classifying retinal diseases from fundus images. The architecture combines two components, Deep Intermediate Feature Extraction (DIFE) and Pointwise Feature Map Reduction (PFMR),built upon a NASNetMobile backbone. The authors demonstrate improved accuracy (90.33%) compared to 14 baseline models while maintaining a compact model size (19.87 MB). While the work addresses an important problem in deploying diagnostic tools in resource-constrained settings, several methodological and experimental concerns should be addressed before publication.

Major Concerns

1. The authors present DIFE and PFMR as novel contributions. However, these techniques are not new in themselves. Extracting intermediate features from CNNs is a well-established practice in transfer learning, and PFMR is essentially a 1×1 convolution for channel reduction, a foundational technique dating back to Network-in-Network (2014) and used extensively in Inception, ResNet bottlenecks, and MobileNet architectures. The manuscript should more transparently acknowledge that the contribution lies in the specific combination and configuration of these existing techniques for retinal disease classification, rather than presenting them as methodological innovations.

2. The manuscript lacks essential reproducibility components. The authors should provide the model implementation code and test scripts, ideally through a public repository. This is particularly important given the claims of superior performance and the specific architectural choices involved in DIFE and PFMR.

3. Training curves are only provided for the proposed LiteFeatNet model. The manuscript does not demonstrate that all comparison models reached training convergence. This is a critical concern, as Table 5 reveals that the EfficientNet models (B0 and V2B0) exhibit pathological behavior, achieving 0% precision and 0% sensitivity for DR and MH classes while predicting NL for all samples. This pattern strongly suggests training failure rather than genuine model limitations. The authors must ensure that all models are correctly and fully trained until convergence before drawing comparative conclusions. The current results for EfficientNet models should either be corrected through proper training or excluded from the comparison with appropriate justification.

4. The authors utilize only 3 of the 46 available disease categories in the RFMiD dataset (DR, MH, and Normal). Several questions arise:

- What criteria drove the selection of these specific categories?

- Can we reasonably expect the DIFE and PFMR approach to generalize to the remaining 43 disease categories, or to a larger multi-class problem?

The limited scope constrains the claims of clinical utility and generalizability.

5. The authors place considerable emphasis on training time as a metric of efficiency (e.g., "6.82 minutes of training"). However, training computational cost has minimal bearing on deployment feasibility in resource-constrained environments. Models are trained once, typically on capable hardware, making it acceptable, even advisable, to invest substantial time at this stage. The relevant metrics for deployment are model size and inference-time computational cost. The manuscript would benefit from refocusing this discussion accordingly.

6. The authors describe a 60:40 train-evaluation split but report that evaluation was performed on a test set of 362 images. The manuscript should clarify the exact composition of the data splits. Specifically, the authors should confirm that: (a) the validation set comprises only the 368 images mentioned, (b) the test set of 362 images was held out and not used during model selection or hyperparameter tuning, and (c) there is no data leakage between splits.

Minor Concerns

1. The ablation study convincingly demonstrates the synergistic benefit of DIFE and PFMR. However, the authors only evaluate these components with the NASNetMobile backbone. Given the promising results, did the authors consider implementing DIFE and PFMR with other backbone architectures (e.g., MobileNetV2, DenseNet121)? This would strengthen claims about the general applicability of the proposed approach and would be a natural extension given the ablation findings.

2. The manuscript frequently references deployment in "resource-constrained environments" without quantitative specification. The authors should provide:

- Concrete examples of target deployment platforms (specific smartphone models, embedded systems, etc.)

- Memory and computational constraints of these platforms

- Threshold model sizes that these environments can reasonably host

This would help readers assess whether the 19.87 MB model size represents a meaningful improvement for practical deployment scenarios.

3. DenseNet121 achieves comparable evaluation performance to LiteFeatNet (89.23% vs. 90.33% accuracy), with the proposed model's primary advantage being approximately 30% smaller size (19.87 MB vs. 26.86 MB). The authors should more carefully contextualize this trade-off and discuss scenarios where this size reduction provides meaningful practical benefit.

Minor errors:

Line 28: "using 1824 images from to three distinct class labels" , delete "to" (should be "from three distinct class labels")

Line 57: "Cost-effective, non-invasive", missing space after comma

Line 95: "cost-effective, scalable", missing space after comma

Line 94: "implementated", should be "implemented"

Line 175: "Theinclusion", missing space (should be "The inclusion")

Line 212: "92.40%accuracy", missing space (should be "92.40% accuracy")

Line 358: "information,and fully leverages" , missing space after comma

Line 521: "Parameters (in numbers), size (in MBs)The LiteFeatNet", missing space or period before "The"

Line 854: "94.04%)while requiring", missing space before "while"

Recommendation

The manuscript presents an interesting architectural combination, and the ablation study provides convincing evidence of the synergistic utility of DIFE and PFMR. However, the experimental evaluation requires significant revision before the conclusions can be considered reliable.

I recommend major revision before this manuscript can be considered for publication.

7. PLOS authors have the option to publish the peer review history of their article (what does this mean?). If published, this will include your full peer review and any attached files.

Reviewer #2: No

Reviewer #3: No

---

## [Author Response · Author response to Decision Letter 2]

19 Mar 2026

Editor:

====

We would like to cordially thank the esteemed editor for providing us with guidelines. We carefully reviewed all the comments from the reviewers. No specific recommendations for citing previously published works were made by the reviewers. As a result, no additional references / citations were added in this revised manuscript upon recommendations. We have carefully reviewed the entire reference list to ensure that all citations are complete, accurate, and up to date. None of the cited articles have been retracted. Therefore, no references required removal or replacement in this regard.

However, while addressing reviewer’s comments during the revision process, several new references were incorporated in the manuscript to contextualize and strengthen our work. We have also cross-checked all the references and to ensure completeness and consistency in formatting according to the journal’s guidelines.

REVIEWER # 2:

We sincerely thank the reviewer for the positive evaluation, encouraging feedback, and kind remarks on our manuscript. We are also pleased and appreciate the reviewer’s recognition of the quality of our work.

REVIEWER # 3:

We thank and appreciate the reviewer for the careful evaluation of our manuscript and for insightful feedback, comments, and suggestions. These remarks have helped us improve the clarity and quality of our manuscript. Our point-by-point response is given below:

Major Concern # 1

We are extremely grateful to the reviewer for highlighting this important concern regarding the DIFE and PFMR architectural components. We also fully agree that the components integrated in our proposed architecture, such as intermediate feature extraction and 1x1 convolution layer, are well-established techniques in deep learning, as used in Inception, ResNet bottlenecks, and MobileNet. However, our intention was not to claim novelty in these operations. Rather, these components were deliberatively integrated to reduce computational complexity, inference time, and model parameters while preserving discriminative feature representations.

Specifically, in DIFE, feature maps are extracted from the NASNetMobile backbone prior to the ReLU activation layer. This strategy allowed the model to utilize feature-rich intermediate representations (7x7x1056) while avoiding additional computation by ReLU. In PFMR, a 1x1 convolutional operation reduces the channel-dimension of DIFE-extracted features to 7x7x512 and a ReLU activation is performed to the reduced feature maps. This further lowers the computational burden of subsequent operations in the classification module. To avoid any ambiguity, we have carefully revised our manuscript. We have clarified that the actual contribution of this work lies in task-oriented integration and configuration of these operations within a unified architecture for retinal disease classification, rather than presenting them as methodological innovations. (Ammendments are mentioned in the Response to Reviewers file)

Major Concern # 2:

We appreciate reviewer’s observation regarding the reproducibility component. The complete code has been made publicly available on the GitHub repository. The link to this repository is provided in the Data Availability Statement under Code Availability section of the manuscript.

Major Concern # 3:

We are extremely grateful to the reviewer for raising this important concern regarding training convergence for providing a fair comparison as base for analysis. Following the kind suggestion of the reviewer, we have carefully reviewed the accuracy and loss curves for all the models. According to these curves, all the models demonstrated consistent stabilization, thereby confirming that all the models were trained until convergence. For clarity, we have also added these curves (for all the architectures used in this study) and related discussion in section 4.2 of this revised version of the manuscript. Accuracy curves of all the models are combined into a single figure Fig. 7 while loss curves are also combined together and presented in a single figure Fig. 8. The figure captions are also updated accordingly.

We have also performed a number of additional experiments according to the valuable suggestions of the reviewer and we truly believe that these suggestions have improved the quality of our manuscript greatly. Following these suggestions, we have added an entirely new section (section 6) and a sub-section (section 7.2) in this revised manuscript in response to major concern # 4 and minor concern # 1, respectively. Therefore, to manage the overall sense, consistency, and length of our manuscript, we have excluded the least performing models from this revised manuscript (including related table entries and corresponding discussion). The revised manuscript contains results and discussion for the baseline models that demonstrated stable convergence and training. We believe that these amendments significantly strengthen the scientific rigor, methodological validity, and fairness of comparison. (The details of ammendments are mentioned in Response to Reviewers file)

Major Concern # 4:

a) Specific Criteria for Selection of these specific categoris:

We sincerely thank the reviewer for careful reading of our manuscript and for raising this insightful question regarding the selection of categories. We also appreciate the reviewer for providing us with the opportunity to clarify our rationale. We confirm that the choice of disease categories was not arbitrary. Instead, our choice was fact-driven, logical, and scientifically valid. These facts are stated below:

RFMiD exhibits substantial class imbalance with several class labels having only a very few samples (below 50 and even below 10). The selected categories for our study: Diabetic Retinopathy (DR), Media Haze (MH), and Normal (NL) are the well-represented classes with having sufficient number of images across the training, validation, and testing splits. The diseases such as DR and MH also have high clinical relevance and practical significance. DR is progressive and leading cause of vision loss worldwide affecting a high number of patients worldwide. Developing countries often face challenges such as limited availability of experienced ophthalmologists and lower diagnosis time per patient for managing the ever-increasing patient count, sometimes leading to misdiagnosis. This sets stage for accurate and reliable automated disease identification. MH is also common in retinal imaging and results in media opacities, thus poses a challenge for automated disease identification from fundus images. So, we selected it to ensure the robustness of our model to variations in image quality. NL images were included for enabling model to distinguish between healthy retinal structures from diseased pathologies. Therefore, we selected these categories due to the following criteria to assess the effectiveness of our proposed methodology.

1. To have clinical and practical significance.

2. To achieve stable model training backed with adequate training samples.

3. To enable reliable performance evaluation.

4. To ensure model’s effectiveness in handling images with varying quality.

We believe that our clarification regarding the choice of category selection strengthens clinical relevance, methodological rigor, transparency, and overall robustness of our study.

b) Expectation to generalize to larger multi-class scenarios:

We are grateful to the reviewer for this important observation regarding the generalization and scalability of our proposed framework, particularly for a larger multi-class problem. Therefore, to evaluate this, we conducted additional experiments using two more common retinal pathologies: Macular Scar (MS) and Myopia (MYA) from the original RFMiD. To overcome severe class imbalance, we supplemented images from another publicly available multi-class retinal dataset. Experimental evaluation was performed using two different scenarios:

Scenario # 1: 4-class labels

Scenario # 2: 5-class labels

Our proposed framework demonstrated a strong and stable performance (accuracy: 88.86% and 85.42% for 4- and 5-class scenarios, respectively) for both experimental scenarios. To provide detailed insights, we also compared its performance with several baseline architectures. These baseline models showed notable performance degradation (particularly from 4-class scenario to 5-class scenario) whereas our proposed model only exhibited a modest performance drop. Importantly, our proposed model achieved competitive performance and maintained significantly lower inference time.

These results provide empirical evidence that our proposed framework is not limited to only a small subset of disease (i.e. DR, MH, and NL from RFMiD). It is robust enough in effectively generalizing to 4-class and 5-class scenarios as well. Therefore, we can reasonably expect the DIFE and PFMR approach to generalize to larger multi-class classification problems.

We have added a new section (6. Scalability assessment) in our manuscript containing all the necessary experimental protocol followed, results, and discussion. Amendments are also made to various other sections of the manuscript due to this valuable addition in our manuscript.

(Ammendments made are clearly described in Response to Reviewers file)

Major Concern # 5:

We cordially thank the reviewer for this insightful observation. We fully agree with training time’s limited relevancy with the real-time deployment and operations, as models are typically trained using capable hardware resources. Therefore, we have updated the manuscript and placed more emphasis on the metrics such as model parameters, size and inference time, which are relevant ones for deployment. Various sections / sub-sections have been updated for refocusing our discussion towards this important point. (Ammendments made are clearly described in Response to Reviewers file)

Major Concern # 6:

We appreciate reviewer for observing and pointing out this important point. We also truly believe that the there is a stronger need to clarify the exact composition of the dataset splits to improve quality and understanding of the manuscript. To clarify and explain this, we have explicitly included the term “train-validate-test ratio” instead of “train-test ratio” or “train-evaluate ratio” in numerous sections of this revised manuscript. Additionally, for absolute clarity, we have also mentioned “60:20:20” in place of “60:40” as “train-validate-test ratio” in this revised manuscript at two distinct locations Abstract and Section 3.1 Dataset description. The stats of the dataset composition are also included in the manuscript as Fig 3. For clarity, we have also amended the caption from “Class label distribution in the RFMiD.” to “Class label distribution for the study.”

We are also grateful to the reviewer for highlighting the need for clarification of various aspects related to dataset splits, images in the splits, and data leakage. We fully agree that the including the confirmation of validation split, usage of test split, and data leakage in the manuscript will truly improve the clarity of the methodology used in this study. (Ammendments made are clearly described in Response to Reviewers file)

a) Validation set images

We truly acknowledge the need for the conformation and clarity regarding the images in the validation set. The validation set was utilized in this study comprises of only 368 images belonging to the chosen class labels. To further clarify this, we have explicitly included the details at multiple locations in this revised manuscript text. We have also explicitly included class-label-wise distribution of image samples across the training, validation, and testing splits in Fig 3. (Ammendments made are clearly described in Response to Reviewers file)

b) Holding test images:

We appreciate the reviewer for this valuable suggestion regarding the test set images. We utilized three independent dataset splits (training, validation, and test) in this study. The test set (with 362 images) was strictly held out during the model development. It was not used for model selection, hyper parameter tuning, or any training-related procedure. We only used the test set for testing/final evaluation of the performance. To provide factual details to readers we have made a number of amendments is this revised manuscript. (Ammendments made are clearly described in Response to Reviewers file)

c) Data Leakage between Splits:

We are grateful to the reviewer for raising this important concern regarding the dataset composition and experimental protocol. We apologize for the lack of clarity in the submitted manuscript and we also believe that providing details on data leakage concern exclusively will truly improve the clarity in our methodology.

The dataset used in this study was already divided into three splits: training (60%, 1920 images), validation (20%, 640 images), and testing (20%, 640 images) splits. According to the dataset descriptor, the original contributors have clearly mentioned that disease-wise stratification was maintained across these splits. In our study, we strictly adhered to these partitions and did not perform any structural changes or re-splitting to the dataset. All the images used in this study were extracted directly from the pre-defined splits while preserving the original preservation. Therefore, we confirm that there was no data leakage between these splits, as we ensured that original splitting, as defined by original contributors, was followed for experiments. (Ammendments made are clearly described in Response to Reviewers file)

Minor Concern # 1:

We appreciate the reviewer for recognizing the quality of the ablation study and pointing out an important concern regarding integration of these modules with other backbone architectures. We also like to thank reviewer for providing us with valuable suggestions and opportunities that will surely strengthen claims of the general applicability of our proposed approach by extending the ablation findings. To do this, we have conducted a detailed ablation study by using 8 different ablation models, utilizing DenseNet121 and MobileNetV2 architectures. The results assure the effectiveness and synergistic benefit of DIFE and PFMR for these architectures as well. We have also added a new sub-section (sub-section 7.2) in this revised manuscript to explain experimental protocol followed, results, and related discussion. (Ammendments made are clearly described in Response to Reviewers file)

Minor Concern # 2:

We are really thankful to the reviewer for keenly observing and reviewing our manuscript, and raising this important concern. We also believe that updating our manuscript in the light of valuable suggestions by reviewer will represent a meaningful improvement for practical deployment scenarios. Therefore providing concrete examples of target platforms, constraints, and thresholds will truly improve the clarity and quality of our manuscript. In response to this concern, we have made a number of changes in this revised manuscript. (Ammendments made are clearly described in Response to Reviewers file)

Minor Concern # 3:

We are grateful to the reviewer for raising this thoughtful concern and valuable suggestion. We fully agree with the reviewer that contextualizing this tradeoff related to model size and performance will greatly benefit our manuscript. Moreover, discussing the practical benefits provided by this reduced size along with real life scenarios will improve the clarity and quality of our manuscript. Therefore, we have updated our manuscript for better contextualization of this trade-off, particularly in connection with the DenseNet121 architecture. (Ammendments made are clearly described in Response to Reviewers file)

Minor Errors

--------------

We are grateful to the reviewer for highlighting various language issues, spelling mistakes, and formatting errors. To improve the clarity and overall quality of

---

## [Editor Report · Decision Letter 2]

8 Apr 2026

LiteFeatNet: A parameter-efficient and performance-centric deep learning model for multi-ocular disease identification using intermediate feature reduction from fundus images

PONE-D-25-44797R2

Dear Dr. RAFI,

We’re pleased to inform you that your manuscript has been judged scientifically suitable for publication and will be formally accepted for publication once it meets all outstanding technical requirements.

The manuscript has been substantially improved following revision, and the authors have addressed most of the reviewer’s concerns in a satisfactory manner. The additional experiments and clarifications provided have strengthened the overall quality and rigor of the study.   However, several minor but important issues remain, which should be addressed prior to final publication. These can be handled during the galley proof stage:

The claims regarding methodological novelty should be carefully toned down to avoid potential overstatement. The contribution should be more clearly positioned as an effective integration and configuration of existing components rather than as a fundamentally novel architectural innovation.The code availability link provided in the manuscript should be re-checked, as it does not appear to be accessible. The authors are requested to verify that the link is functional and publicly available.A final round of English language polishing is recommended to improve clarity, grammar, and overall readability of the manuscript.

Subject to these minor revisions, the manuscript is suitable for publication.

Kind regards,

Taikyeong Ted Jeong, Ph.D.

Academic Editor

PLOS One

Additional Editor Comments (optional):

The manuscript has been substantially improved following revision, and the authors have addressed most of the reviewer’s concerns in a satisfactory manner. The additional experiments and clarifications provided have strengthened the overall quality and rigor of the study.

However, several minor but important issues remain, which should be addressed prior to final publication. These can be handled during the galley proof stage:

The claims regarding methodological novelty should be carefully toned down to avoid potential overstatement. The contribution should be more clearly positioned as an effective integration and configuration of existing components rather than as a fundamentally novel architectural innovation.

The code availability link provided in the manuscript should be re-checked, as it does not appear to be accessible. The authors are requested to verify that the link is functional and publicly available.

A final round of English language polishing is recommended to improve clarity, grammar, and overall readability of the manuscript.

Subject to these minor revisions, the manuscript is suitable for publication.
---

## [Editor Report · Acceptance letter]

PONE-D-25-44797R2

PLOS One

Dear Dr. RAFI,

I'm pleased to inform you that your manuscript has been deemed suitable for publication in PLOS One. Congratulations! Your manuscript is now being handed over to our production team.

Kind regards,

on behalf of

Professor Taikyeong Ted Jeong

Academic Editor

PLOS One